# Mechanofusion-derived cathode composite microstructures with scalable mixed conducting matrix coatings for solid state batteries

Maximilian Kissel [1,4], Finn Frankenberg [2,4], Thomas Demuth [3], Anton Lai [1], Niklas Laser[2], Daniel Wagner [1], Ahmed Eisa[2], Peter Michalowski [2], Kerstin Volz[3], Arno Kwade [2] ✉ & Jürgen Janek [1] ✉

The successful implementation of solid state batteries not only requires the use of high-capacity anodes, but also high-performance composite cathodes. However, the production of solid state battery cathode composites with optimized microstructures remains a significant challenge, especially for large-scale fabrication. Here, we present a scalable high-intensity dry mixing process to create tailored functional coatings on single-crystalline $LiNi_{0.82}Mn_{0.07}Co_{0.11}O_2$ via mechanofusion. We investigate the coating of $LiNi_{0.82}Mn_{0.07}Co_{0.11}O_2$ with the malleable halide solid electrolyte $Li_3InCl_6$ under various process conditions, linking process parameters obtained from discrete element method simulations with experimentally accessible morphological properties to offer guidelines for further optimization. In this way nanometer-thin covering coatings as well as thick matrix coatings are successfully produced. Incorporating carbon black into the thick matrix coating results in well-performing mixed conducting matrices that can be used directly as composite cathodes without further treatment. The compositions investigated enable stable cycling with a specific capacity of up to $q_{comp} = 100\ mAh\ g^{-1}$ (based on the total mass of the composite cathode) at a C-rate of 1 C (60 min). While higher carbon black content is observed to improve CAM utilization, excessive amounts are detrimental for cell kinetics and chemo-mechanics, emphasizing the importance of the cathode mixing process and composition on overall cell performance.

Solid state batteries (SSBs) have emerged as a promising next-generation energy storage technology, aiming to overcome the safety and energy density limitations of conventional lithium-ion batteries (LIBs) that rely on liquid electrolytes (LEs)[1,2]. Replacing the LE with a solid electrolyte (SE) is expected to enable the implementation of high-energy anode (negative electrode) concepts such as lithium metal anodes[3], "anode-free" configurations[4], or Si-based anodes[5]. To match these expected high-performance anodes, equally high-performing cathodes (positive electrode) are required[6–8].

[1]Institute of Physical Chemistry & Center for Materials Research, Justus-Liebig-Universität Gießen, Gießen, Germany. [2]Institute for Particle Technology, Technische Universität Braunschweig, Braunschweig, Germany. [3]Materials Science Center (WZMW) and Department of Physics, Philipps-University Marburg, Marburg, Germany. [4]These authors contributed equally: Maximilian Kissel, Finn Frankenberg. ✉e-mail: arno.kwade@tu-braunschweig.de; juergen.janek@uni-giessen.de

SSB cathodes are usually designed as composites of SE, cathode active material (CAM), and additive particles. Towards large-scale commercialization of SSBs, various challenges persist, starting at the material level[9]. Here, research focuses on the one hand on developing advanced SEs with enhanced ionic conductivities[10,11], broader electrochemical stability windows[12,13], or beneficial mechanical properties[14] and on the other hand scaling and sustainable SE production processes[15,16]. Moreover, to mitigate degradation effects at the CAM|SE interface, the use of protective CAM coatings is required[17,18]. As CAM, Ni-rich layered oxides[19–21] are widely used with preference for materials that show beneficial chemo-mechanical properties, i.e., low volume expansion[22–24] and low tendency for cracking[25,26]—as seen in single crystals, which showed improved long-term cycling performance[9,27–30].

Not only the materials in the cathode but especially their arrangement, i.e. the composite microstructure, is crucial for the performance[7,31–33]. The microstructure needs to fulfill three main tasks: First, the CAM particles need to be electronically and ionically connected, i.e., electrochemically active, leading to a high CAM utilization within a percolating network. As recently highlighted by Kissel et al.[34], achieving a complete static CAM utilization is not self-evident in SSBs. Second, the electronic and ionic networks should possess high effective conductivities for an improved kinetic cell performance[35–38]. Lastly, intimate and mechanically as well as (electro)chemically stable solid-solid contacts between the CAM and the SE are crucial to enable fast charge transfer at the interface, while also accommodating the volume expansion of the CAM during charging and discharging[39,40].

The final SSB cathode microstructure is largely defined during the particle mixing process, since, in contrast to LIBs, SSBs usually do not allow infiltration of the cathode with an electrolyte post-assembly. However, despite its importance and challenges toward large-scale implementation[41–43], the actual mixing process has received relatively little attention up to now[44–47]. In academic studies, hand mortaring is common, although this is non-scalable and suffers from reproducibility issues[34,48]. On the other hand, machine-made composites are becoming more prevalent in recent studies[34,43,49,50]. Further processing-related approaches have focused on comminuting SE particles before mixing to achieve a tailored particle size distribution (PSD)[49,51–54]. While this is crucial to match the CAM PSD[7,38,52] it involves additional processing steps that can damage the SEs by solvents[55] or by the milling itself[49,56]. Other approaches are based on solution processes to apply SE coatings on active materials or to infiltrate porous electrodes[57,58]. In general, only a few studies explore scalable composite mixing machines for SSB cathode composites, and in most cases, the corresponding mixing process leads to a more or less "random" ordering of particles[7,43,47].

Recently, employing a combined modeling and experimental approach, Lee et al.[59] introduced the concept of building blocks towards an ideal cathode composite microstructure. Building blocks describe the fundamental units of a composite's microstructure in the form of tailored particle aggregates. By engineering these building blocks, the overall microstructure and thus the resulting material properties can be precisely controlled. Lee et al.[40] highlighted the importance of shear forces, which increase the electrolyte-covered surface area of the active materials. Kawaguchi et al.[60,61] and Hayakawa et al.[62,63] investigated the idea of coating low- and medium-nickel $Li_xNi_yCo_zMn_{1-y-z}O_2$ (NCM) particles with SE in a high-intensity mixer. Kim et al.[39] utilized mechanofusion as a preprocessing step to ensure intimate interfacial contact by coating NCM with $Li_6PS_5Cl$ (LPSCl). This allowed a high CAM loading in the cathode and led to higher specific capacities. However, sole coating with SE is not sufficient since it impedes electron transfer, which would only be possible via CAM-CAM particle contacts after deforming parts of the SE coating layer during densification. To overcome this limitation, the authors added carbon additives to their SE-coated CAM and had to manually mortar the composite once again[39].

Inspired by these recent works, we systematically investigate a dry particle coating approach, based on mechanofusion, in this study to create tailored building blocks for SSB composite cathode microstructures. Going beyond the idea of simple mixing of loose particles and arbitrary (re-)ordering, we employ a scalable high-intensity mixing process to create a defined mixed conducting matrix coating via mechanofusion on single-crystalline Ni-rich $LiNi_{0.82}Mn_{0.07}Co_{0.11}O_2$ (NCM82) without any hand mortaring being involved. In this bottom-up approach, which can also be used to produce protective CAM coatings, we make use of the good deformability of the halide SE $Li_3InCl_6$ (LIC) as a model catholyte. We investigate the effect of composition and process parameters on the morphology of the coating, coating progress, and possible degradation of the materials, combining scanning electron microscopy (SEM), energy-dispersive X-ray spectroscopy (EDX), (scanning) transmission electron microscopy ((S)TEM), and X-ray photoelectron spectroscopy (XPS). We link experimental results with stressing conditions obtained from discrete element method (DEM) process simulations to describe the coating process on the macroscale. In this way, we establish a basis for future optimization of the mixing process of composite cathodes. We demonstrate that the presented mechanofusion process is suitable to produce thin covering coatings as well as thick matrix coatings. By adding carbon black (CB) into the matrix, the coatings become mixed conducting and are further electrochemically analyzed in detail. Our results show that the mixed conducting matrix coating design approach via mechanofusion is promising as a versatile and scalable production method for SSB composites, while still offering significant potential for further optimization.

## Results

Figure 1 illustrates the concept of the mixed conducting matrix coating approach employed in this study. Cathode composites, comprising three components with different PSDs, were fabricated using single-crystalline NCM82, in the following denoted as NCM, with a median particle size of $d_{50} = 3.3\,\mu m$ as CAM, in-house prepared LIC (see "Methods") with particle sizes ranging from 100 nm to 100 μm as SE and CB ($d_{50,aggregates} \approx 250\,nm$) as electron-conducting additive (Fig. 1a).

The particles were mixed in a scalable and commercially available high-intensity mixer (Fig. 1b), which provides high-shear forces within the narrow gap[43], enabling mechanofusion[39,45,64,65]. During this process, the NCM particles act as host particles that are coated with guest particles (here: LIC or premix of LIC and CB, see Table 1 and "Methods"), forming core-shell type heteroaggregates as depicted in Fig. 1c. These aggregates are hereafter referred to as building blocks, following the concept introduced by Lee et al.[59] and are denoted by their composition as NCM:LIC:CB (w/w/w). Process parameters that can be varied during the high-intensity mixing are the rotational speed $n$, the mixing time $t_{mix}$, and the filling degree $\varphi$. By adjusting these parameters, the properties of the building blocks in this bottom-up approach, and thus the overall microstructure of the cathode composite (Fig. 1d), can be significantly influenced. Towards scalable cathode composite production, all mixtures in this work were prepared in a dry room with a dew point of the supply air of −60 °C without any manual hand mortaring being involved. All mixtures were prepared with a batch size of approximately 20 g, corresponding to a filling degree of 10%. In total, three main investigations (cf. Tables 1 and S1) were conducted and are discussed in the following.

First, the coating thickness of the building blocks was varied by systematically adjusting the ratio of SE (and CB) to NCM, while maintaining constant process time and rotational speed. Secondly, the influence of stress intensity or shear stress, as well as collision number,

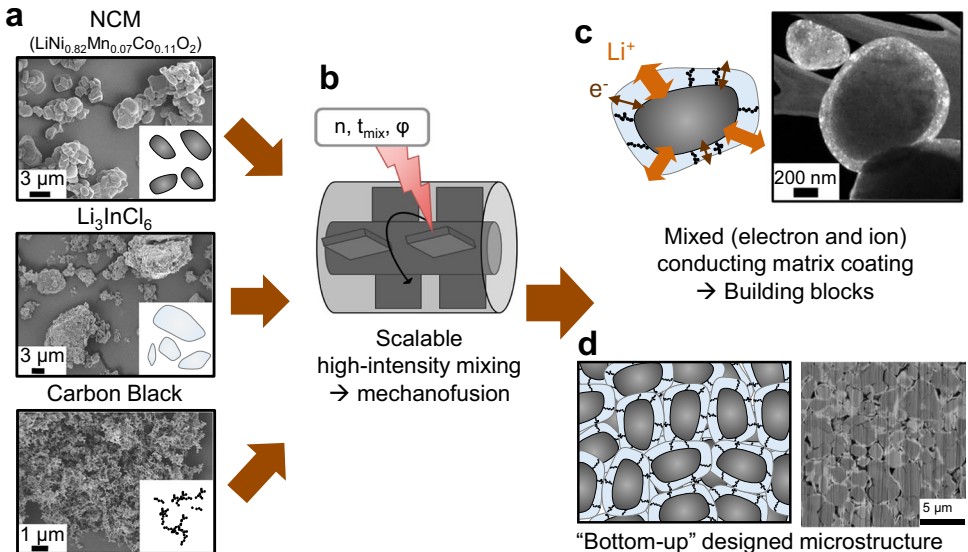

**Fig. 1 | Concept of the mixed conducting matrix coating approach. a** Starting materials with different particle size distributions are **b** mixed in a high-intensity mixer with varying rotational speed $n$, mixing times $t_{mix}$, compositions, and constant filling degree $\varphi$, leading to mechanofusion. **c** NCM host particles are coated with an electron and ion conducting matrix, forming building blocks that build up an ordered composite cathode microstructure (**d**).

**Table 1 | Overview of samples for the different investigations presented in this study**

| Type of investigation | Investigated parameters | Constant parameters | Nominal compositions (NCM:LIC:CB, w/w/w) |
|---|---|---|---|
| Coating thickness variation (without CB) | Coating content variation: 1–20 wt% | -Mixing NCM + LIC: 60 min at 10,000 rpm<br>-Filling degree: 10% | 99:1:0<br>98:2:0<br>95:5:0<br>90:10:0<br>80:20:0 |
| Coating thickness variation (with CB) | Coating content variation: 1–20 wt% | -Fixed CB-LIC ratio (20 vol%)<br>-Premixing CB + LIC: 10 min at 5000 rpm<br>-Mixing Premix + NCM: 60 min at 10,000 rpm<br>-Filling degree: 10% | 99:1:0.2<br>98:2:0.3<br>95:5:0.8<br>90:10:1.5<br>80:20:3 |
| Process parameter variation | -Mixing times: 5, 10, 30, 60 min<br>-Rotational speeds: 1000, 2500, 5000, 7500, 10,000 rpm | -Premixing CB + LIC: 10 min at 5000 rpm<br>-Filling degree: 10% | 95:5:0.8<br>80:20:3 |
| Matrix optimization | CB content variation: 0–3 wt% | -Premixing CB + LIC: 10 min at 5000 rpm<br>-Mixing Premix + NCM: 60 min at 10,000 rpm<br>-Filling degree: 10% | 80:20:0<br>80:20:0.2<br>80:20:0.5<br>80:20:1<br>80:20:2<br>80:20:3 |

respectively, during mixing was investigated by varying the rotational speed and mixing time for two specific compositions. Third, the mixed conducting matrix coating with a weight ratio of 80:20:$x$ (NCM:LIC:CB) was optimized by altering the CB content $x$, aiming at achieving a balance between electronic and ionic partial conductivity within the building blocks.

## Impact of composition on coating morphology

To investigate the effect of coating material content on the morphology and homogeneity of the coating, different mixtures with coating contents ranging from 1 to 20 wt% were processed at a rotational speed of 10,000 rpm, corresponding to a tip speed of $40\,m\,s^{-1}$, for 60 min. As guest particles during the mechanofusion process, pure LIC as well as a LIC-CB mixture with a fixed CB:LIC ratio of 20 vol% CB was used (cf. Table S1). Scanning electron microscopy (SEM) combined with EDX was carried out to visualize the morphology of the coating within the building blocks.

Figure 2a shows that a coating is achieved for all cases, independent of the coating material content. For all EDX acquisitions, the same parameters were used, so that differences in signal intensity can be related to differences in the composition. As expected, the intensity of the chlorine signal, stemming from the coating, decreases with decreasing coating amount. The effect of the coating is further evidenced by the rounding of NCM particle edges, reflected in a decreased aspect ratio and an increased circularity with rising coating material content (Supplementary Fig. S1). This indicates that the coating material fills recessed regions of the NCM particles, leading to a varying thickness across the particle surface (Supplementary Fig. S1e). Intergranular boundaries existing within aggregates of the bare NCM disappear completely for high coating amounts, showing that the coating material is covering these areas (Supplementary Fig. S2). Especially at high coating material contents, an increase in volume-based particle size $Q_3$ is observed, which is based on the increase in coating thickness (Fig. 2b). However, the coating as such is

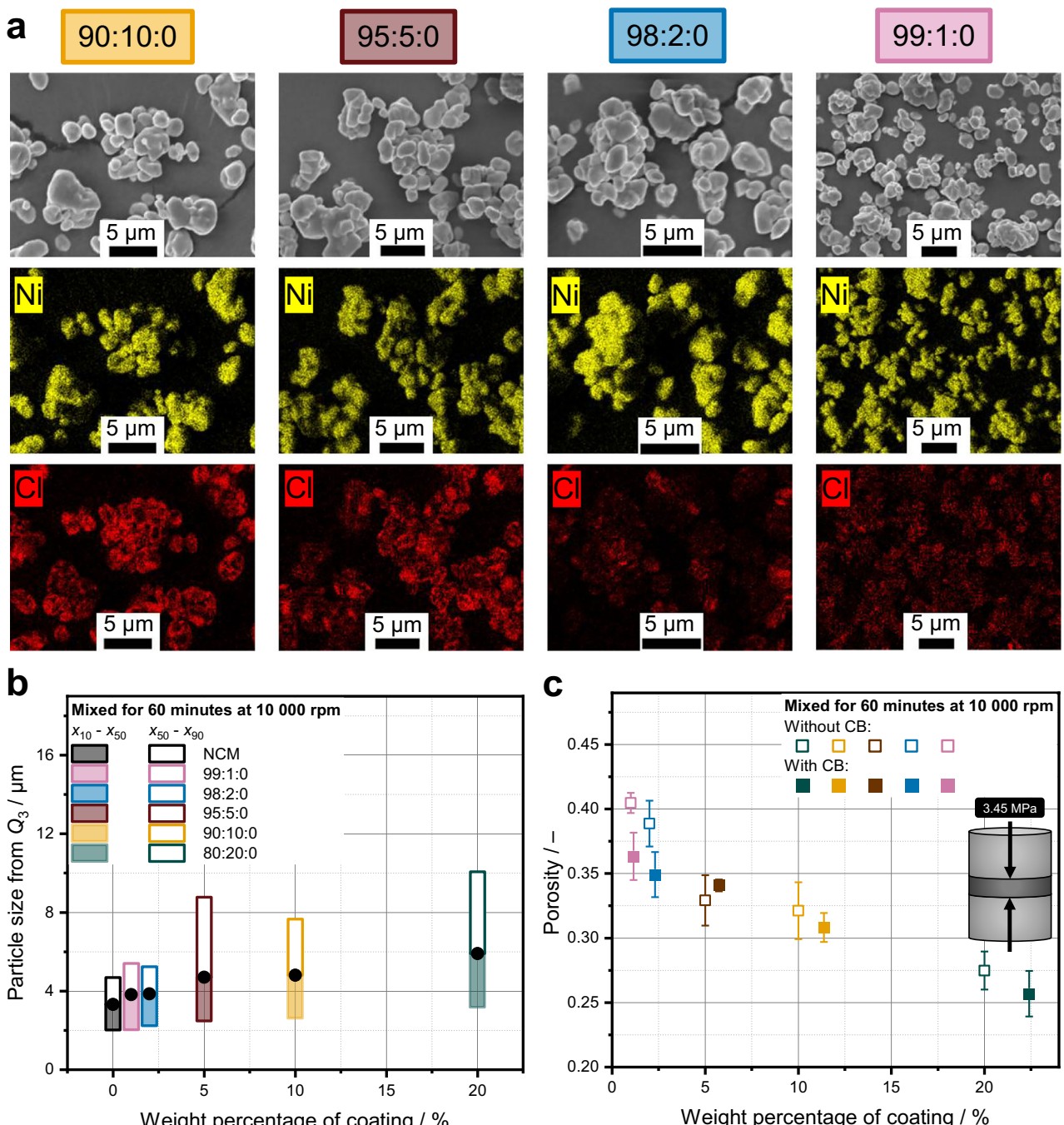

**Fig. 2 | Microstructure, PSD, and porosity depending on the coating amount.** **a** SEM images and EDX maps of four mixtures with decreasing coating amounts. **b** Corresponding PSDs calculated from SEM images segmentation. **c** Porosity of composite pellets measured at 3.45 MPa. Error bars represent the standard deviation of $n = 3$ porosity measurements.

not the determining factor for this increase, since a theoretical maximum average coating thickness of approximately 150 nm is expected (Supplementary Fig. S3). Rather, multiple agglomerated NCM particles are surrounded by a coating layer, which means that one particle is visible under the SEM and counted in the PSD calculation, but in fact several NCM particles are encapsulated by the coating. Since encapsulation occurs only for some agglomerated NCM clusters, a pronounced broadening of the PSD is observed with increasing coating content. From the broadening of the PSD in combination with the increasing amount of soft coating with increasing coating material content, it can be supposed that the interparticle spaces can be better

filled and thus the building blocks can be better packed in the final cathode composite microstructure. This improved packing is indeed reflected in the measured bulk porosity of the cathode composites, which decreases with increasing coating content (Fig. 2c). The porosity of the composite mixtures containing CB is consistently slightly lower than that of the composites without CB. We attribute this to the higher volume fraction of compressible components in the formulations containing CB, which increases from 31.4 to 35.6% for the 80:20 and 80:20:3 compositions, respectively. For the highest coating amount, the porosity measured at a compaction pressure of 3.45 MPa was about 25%. This value corresponds approximately to the porosity of a

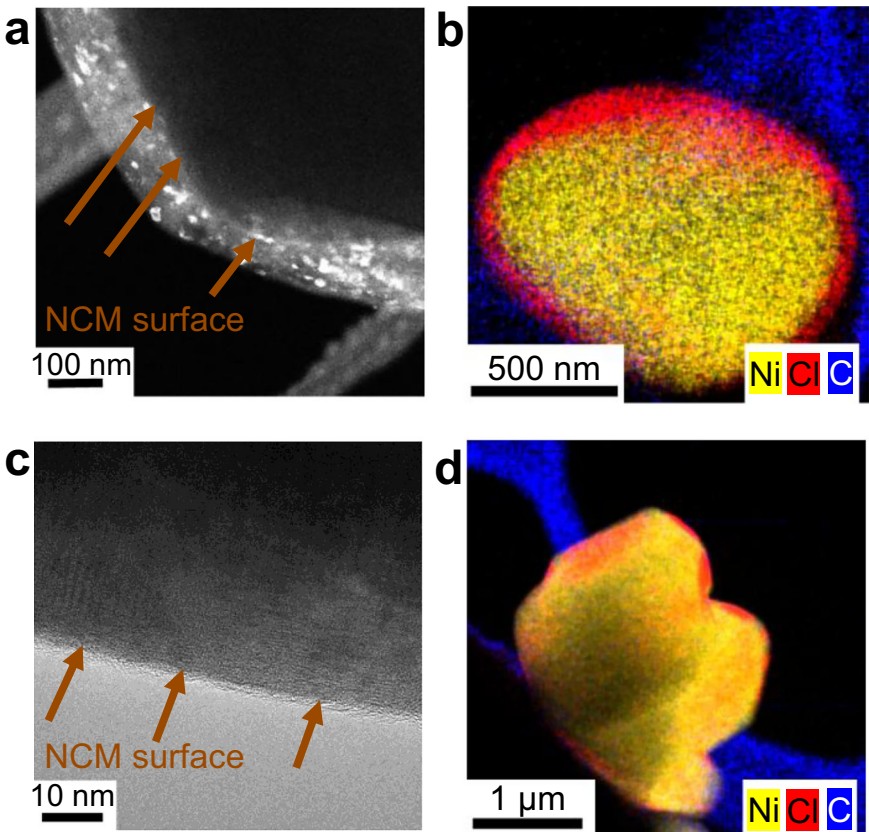

**Fig. 3 | TEM images and STEM-EDX maps of coated particles. a**, **b** Matrix coating building block with composition 80:20:1. **c**, **d** Thin coating with composition of 99:1:0.

cubic closed packing of spheres. For a higher compaction pressure, we expect more deformation of the coatings and thus a reduced porosity of the composite cathode.

Transmission electron microscopy (TEM) images confirm a homogenous coating layer, exemplarily depicted for a high and low coating amount in Fig. 3. In dark-field imaging of a composition of 80:20:1, some crystals in the coating layer appear bright (Fig. 3a). These bright regions are lattice planes of small crystals (cf. Supplementary Fig. S4a) with random orientations towards the position of the objective lens aperture. X-ray diffractograms (Supplementary Fig. S5) confirm that the crystalline phase of the LIC is preserved, and the mass ratios after Rietveld refinement (Supplementary Table S3) are in line with the expected values, indicating that there is no significant amorphization happening. Corresponding STEM-EDX maps (Fig. 3b and Supplementary Fig. S4b) reveal the presence of carbon in the approximately 100 nm thick coating layer, indicating that the desired mixed conducting matrix coating was achieved. In the case of the 99:1:0 composition, the coating is only approximately 5 nm thick (Fig. 3c, d) and consistently observed along the entire particle surface (Supplementary Fig. S6). Variations in signal intensity across the EDX maps in Fig. 3b, d are attributed to differences in particle orientation relative to the detector, which can influence the intensity of the detected signal.

To further quantify the homogeneity and thickness of the coating, XPS was carried out. Based on the signal intensities of the NCM and coating elements, values for the coverage γ of the building blocks were calculated, which represent the signal intensity ratio of coating to NCM (see "Methods" and ref. 66 for details). The coverage value describes how strongly the signal of the NCM particle is shielded by the coating. It depends on the thickness, the composition (with or without CB), the morphology, and the homogeneity of the coating. A coverage value close to 1 indicates nearly complete surface coverage and the presence

of a relatively thick coating layer (cf. Fig. 4a), such that almost no signals from the NCM are detected. This situation corresponds to coating thicknesses of at least ~10 nm, which is approximately the information depth of XPS measurements. While the coverage value itself does not provide information on the morphology or local thickness distribution, it enables assessing whether a continuous coating layer is formed across a broad range of process parameters. For coatings expected to be thinner than ~10 nm (cf. Supplementary Fig. S3), the coverage value derived from XPS no longer reflects the true surface coverage. In this case, even a perfectly covering coating layer will never result in a coverage value of 1, since signals stemming from the NCM can always be detected. Conversely, coatings that are uniformly thicker than at least ~10 nm cannot be distinguished from one another by this method, as all yield coverage values close to 1. As displayed in Fig. 4b, the coverage increases with increasing coating content, which we attribute to the rise in coating thickness. It is also evident that the incorporation of CB in the coating, especially at low coating thicknesses, leads to larger coverage values. Since the weight ratios are kept constant, the coatings that also contain CB occupy a larger volume due to the low (bulk) density of CB. Thus, the signals of the NCM are more strongly shielded, yielding higher coverage values than the CB-free counterparts. At a high coating content corresponding to the ratio 80:20:3, the maximum possible coverage value of 1 is reached, which indicates that the NCM particles are fully covered. This confirms the previously discussed SEM and STEM measurements.

The results demonstrate that high-intensity mixing leading to mechanofusion is a well-suited method for the dry production of core-shell structures with various coating thicknesses. This is particularly attractive for protective coatings employing soft SEs, a promising concept that has recently gained attention to prevent interfacial degradation in SSB cathode composites[65–67].

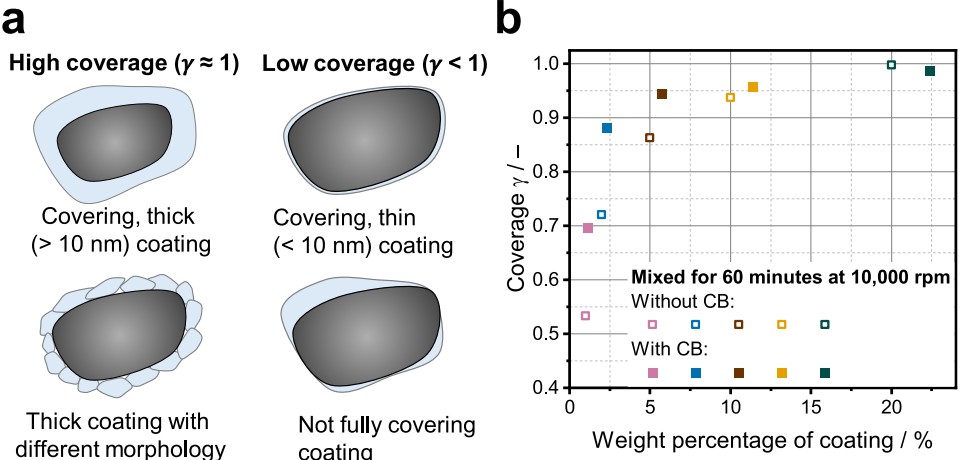

**Fig. 4 | Coverage types and coverage dependence on coating content.**
**a** Schematic illustration of different particle types for which a high and low coverage $\gamma$ can be calculated. **b** XPS-derived coverage values for samples mixed at 10,000 rpm for 60 min, plotted against the weight percentage of coating. Data is shown for composites with and without CB.

## Investigation of process-structure-relations

In the following, we focus on the 80:20:3 composition, which is designed to serve as a matrix coating since it reaches the highest degree of coverage. To investigate the coating progress as a function of process conditions, the composition was produced at different speeds and mixing times.

During high-intensity mixing, mechanical energy is transferred to the particle system through the high relative velocities generated between the rapidly rotating rotor and the surrounding particles. This energy transfer is reflected in the specific energy input $E_{m, exp}$, which is significantly influenced by the rotational speed and mixing time. This was estimated experimentally for the different coating experiments from the measured power $P$, idle power $P_0$, sample mass $m_{total}$, and mixing time $t_{mix}$ (cf. Supplementary Fig. S7).

$$E_{m, exp} = \frac{\int (P - P_0)dt}{m_{total}} \quad (1)$$

To evaluate the coating progression, the coverage of the 80:20:3 composition is depicted in Fig. 5a as a function of the specific energy input. At lower rotational speeds of 1000 rpm and 2500 rpm, coverage increases with prolonged mixing time and thus specific energy input. In contrast, at higher rotational speeds of $n \geq 5000$ rpm, a maximum coverage of 97–100% (dashed gray region) is reached within just 5–10 min of mixing. Extending the mixing time up to 60 min, results in only marginal changes in coverage values at these higher rotational speeds. A similar trend is observed for a composition of 95:5:0.8 (cf. Supplementary Fig. S8a).

As discussed, similar coverage levels do not necessarily indicate comparable coating morphologies (cf. Fig. 4a). Porosity measurements show a consistent decline in porosity of the cathode composites with increasing specific energy input (Fig. 5b). The reduction in porosity indicates that even when coverage values reach a plateau, the morphology of the building blocks can continue to evolve significantly with varying processing conditions. This is corroborated by SEM images as seen in Fig. 5c for the samples processed at different rotational speeds (see also Supplementary Fig. S9). At 1000 rpm, only an inhomogeneous and particulate deposition of CB and LIC on the NCM particles can be seen, forming cluster-like agglomerates. Moreover, single LIC particles are visible, which have not taken part in the coating process. At 5000 rpm, individual LIC and CB

particles remain discernible on the NCM surface, whereas at 10,000 rpm, the NCM particles are uniformly encapsulated by a smooth coating layer. According to mechanofusion theory[68] different coating structures can form at different rotational speeds, ranging from loosely adhered particles to continuous shells. In our case, continuous and smooth shells are observed at high rotational speeds and long mixing times, while lower speeds and reduced energy input tend to result in coatings consisting of loosely adhered particles.

Overall, the coating morphology becomes smoother and less particulate with increasing specific energy input and, at comparable specific energy input, with increasing rotational speed and thus mixer intensity. This leads to an increase in the circularity of the building blocks, as shown for the rotational speeds of 1000 rpm, 5000 rpm and 10,000 rpm mixed for 60 min. Coverage metrics derived from XPS analyses cannot fully resolve these morphological features, which require complementary SEM analysis.

## Linking DEM simulations and experiments

To enable a guided optimization of the mechanofusion process for production of SSB cathode composites, we carried out DEM simulations to simulate the stresses acting on the particles in the high-intensity mixer. As elaborated in the "Methods" section and Supplementary Note 3, a coarse-graining approach was applied to handle the extremely high number of real particles. While this approach does not retain information on the original coating progression, it allows for the determination of the mixer's overall stressing conditions. Calibration tests have been carried out to determine the apparent properties of the coarse-grained particles to mimic the original situation. To validate the calibration procedure and calculated coarse-grain density of the coarse-grained particles (Fig. 6b and Supplementary Fig. S10 and Table S2), the mean specific power obtained from the experiments was first compared with the specific power obtained from the DEM simulation in the steady state. Both specific power data show a high degree of agreement, confirming the successful calibration of the particle properties.

In order to quantitatively describe the process and compare process parameters, we applied the stressing model of Kwade[69], which uses the stress energy or stress intensity, respectively, and stress frequency or stress number, respectively, as parameters to calculate the specific energy input $E_m$ based on these model parameters. This allows a quantitative comparison of various process conditions[70,71]. According

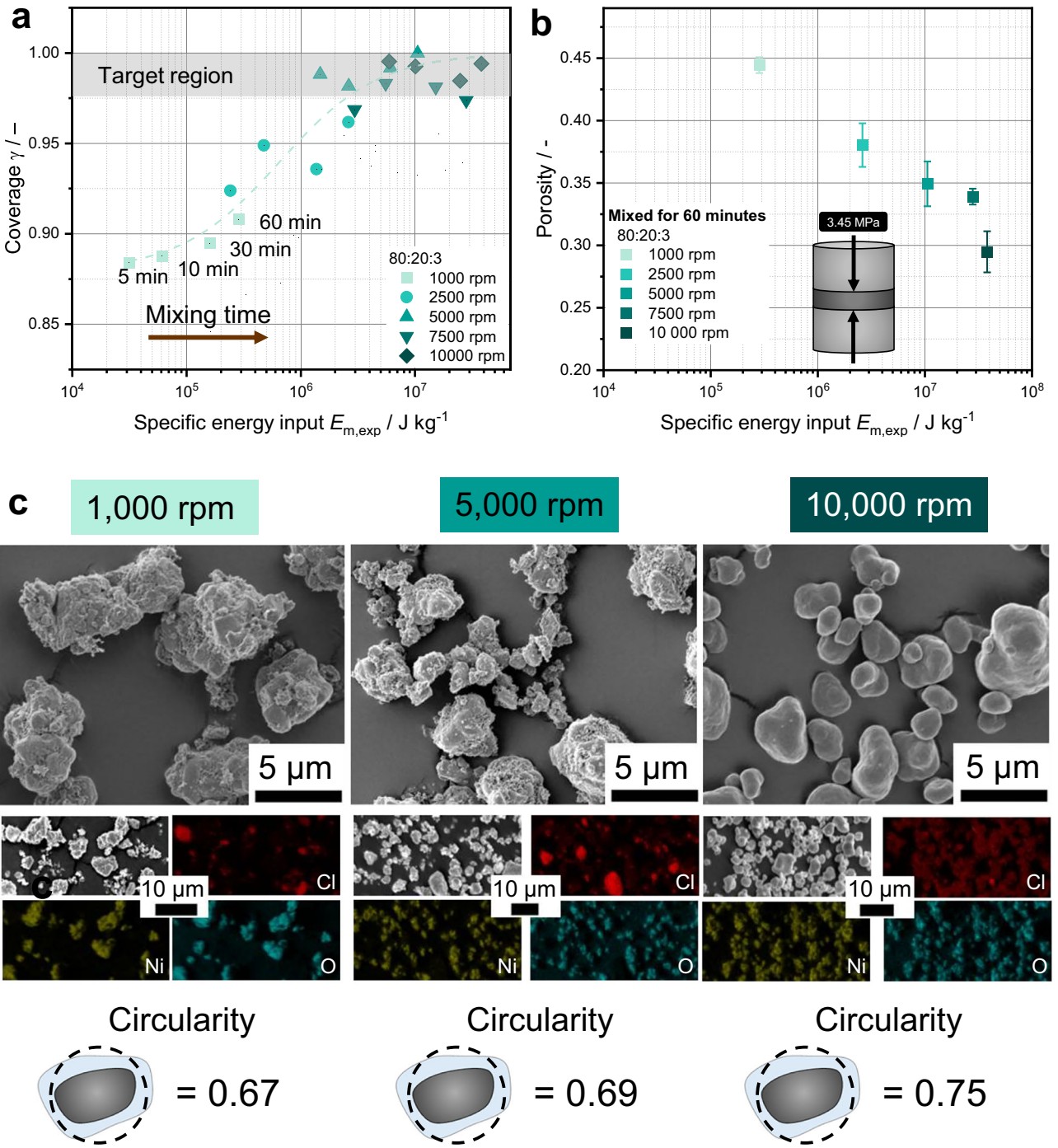

**Fig. 5 | Effect of mixing intensity on particle morphology, surface coverage, and porosity for a building block composition of 80:20:3. a** Coverage γ as a function of specific energy input. **b** Porosity of composite pellets under 3.45 MPa prepared from mixtures after 60 min, plotted against specific energy input. Error bars represent the standard deviation of $n = 3$ measurements. **c** SEM images and EDX maps of powders mixed at 1000, 5000, and 10,000 rpm for 60 min as well as the obtained mean circularity of the different building blocks.

to Eq. 2, $E_{m,sim}$ is expressed as the product of the mean stress intensity $\overline{SI}_{CG}$ of the particle collisions, corresponding to the mean dissipated energy during coarse-grained particle collisions related to stressed coarse-grain mass, and the mean stress number per particle $\overline{SN}_p$. The $\overline{SN}_p$ can be determined by the frequency of collisions of the coarse-grain particles during mixing $SF_{CG}$ divided by the number of coarse-grains, which is obtained from the ratio of the individua particle mass $m_{particle,CG}$ to total mass $m_{total}$, and the mixing time $t_{mix}$. Both parameters $\overline{SI}_{CG}$ and $\overline{SN}_p$ should be approximately scale-invariant

(Supplementary Note 3) and are obtained from DEM simulations[43,72] to calculate the simulation-based $E_{m,sim}$.

$$E_{m,sim} = \frac{\overline{SI}_{CG} \cdot m_{particle,CG} \cdot SF_{CG} \cdot t_{mix}}{m_{total}} = \overline{SI}_{CG} \cdot \overline{SN}_p \quad (2)$$

In the following, the parameters $\overline{SI}_{CG}$ and $\overline{SN}_p$ are used to describe the mechanofusion process macroscopically. Both are individually influenced by the rotational speed $n$ and $t_{mix}$, thus affecting the

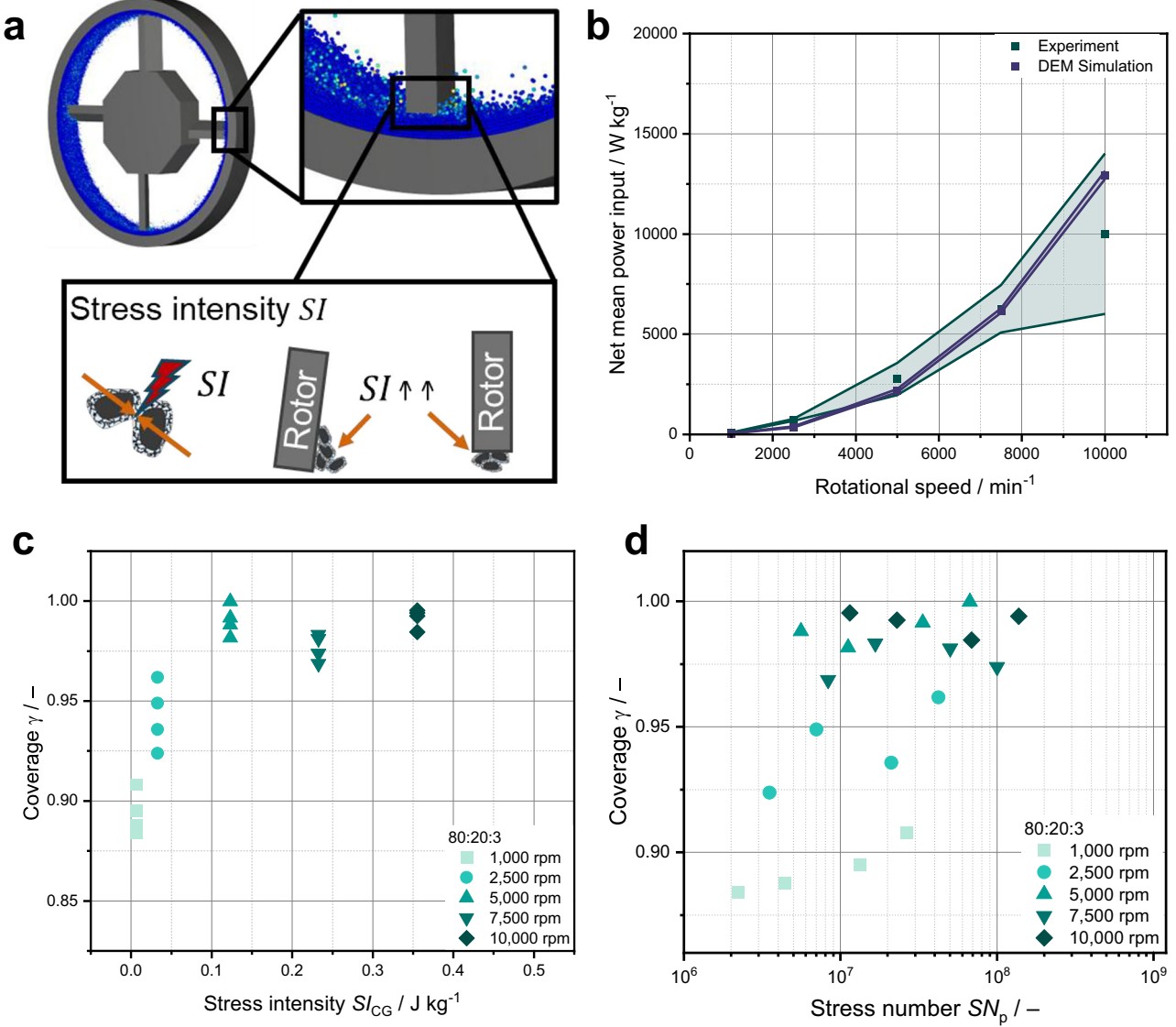

**Fig. 6 | DEM simulation of the high-intensity mixing and correlation with experimental coverage. a** DEM simulation snapshot and schematic image of the high-intensity mixer indicating the mixing action. **b** Measured and simulated net specific power depending on process time for different rotational speeds and a composition of 80:20:3. Error bars represent the standard deviation of the net power, calculated over 60 min for the experiment and over 0.5 s for the DEM simulations. **c** Calculated Coverage as a function of simulated stress intensity $\overline{SI}_{CG}$. **d** Calculated coverage as a function of simulated stress number $\overline{SN}_p$. Regions 1 and 2 correspond to low and high $\overline{SI}_{CG}$, respectively.

progression and morphology of the coating. $\overline{SI}_{CG}$ is highest near the rotor tip and in the gap between the rotor tip and the mixer wall (Fig. 6a), primarily governed by the rotational speed of the mixer for a constant gap width. An approximate increase of $\overline{SI}_{CG} \propto n^2$ can be expected with increased rotational speed[43], potentially influencing the building block microstructure. In contrast, $\overline{SN}_p$ depends on both the rotational speed and $t_{mix}$. Therefore, an equivalent energy input can be achieved at lower $\overline{SI}_{CG}$ if compensated by a higher $\overline{SN}_p$ through extended mixing times, although this may result in a different coating morphology.

To understand the individual influence of $\overline{SI}_{CG}$ and $\overline{SN}_p$, stress frequency–stress intensity curves were generated from the recorded particle collisions and served as the basis for determining $\overline{SN}_p$ and $\overline{SI}_{CG}$. The coverage of the 80:20:3 mixtures was plotted against both parameters (Fig. 6c, d). Figure 6c shows that an increasing $\overline{SI}_{CG}$ generally leads to higher coverage values. Notably, at a $\overline{SI}_{CG}$ corresponding to 5000 rpm ($\overline{SI}_{CG} = 0.123\,\text{J kg}^{-1}$), nearly complete coverage (>97%) is achieved after only 5 min of mixing. Further increasing $\overline{SI}$ beyond this

point results in only slight changes in coverage. However, significant changes in coating morphology that occur at constant mixing time, but varying stress intensities, as already shown in Fig. 5a (see also Supplementary Fig. S9), become visible.

When the coverage is analyzed as a function of $\overline{SN}_p$ (Fig. 6d), it becomes clear that there is no direct proportionality. An equivalent or even higher $\overline{SN}_p$ (region 1) does not achieve the same degree of coverage but depends on the $\overline{SI}_{CG}$. This indicates that elevated $\overline{SI}_{CG}$ (region 2), associated with higher collision energies, is beneficial for forming a uniform coating with higher coverages. Nevertheless, even at lower $\overline{SI}_{CG}$ such as $0.006\,\text{J kg}^{-1}$ (1000 rpm) and $0.033\,\text{J kg}^{-1}$ (2500 rpm), the coverage gradually increases with $\overline{SN}_p$ due to extended mixing times. However, to achieve more than 97% coverage with a $\overline{SI}_{CG}$ equivalent to 2500 rpm and 1000 rpm, longer mixing times would be required. Even then, the resulting coating morphology is likely to differ significantly, as already observed at higher rotational speeds (5000 rpm and 10,000 rpm), where 97% coverage was also achieved (cf. Fig. 5c). From a process engineering and economic perspective,

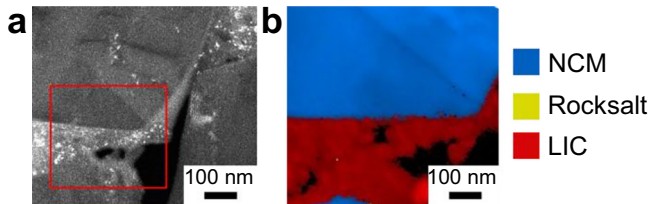

**Fig. 7 | Structural damage probed with SPED-TEM. a** TEM bright-field image of coated NCM (80:20:3) with the red box indicating the area of the SPED scan. **b** SPED phase map showing only the pristine layered phase and the coating layer. Black points represent mostly amorphous regions or voids.

long mixing times are impractical. Therefore, a higher $\overline{SI}_{CG}$ is preferable to ensure efficient coating formation within reasonable processing times.

To achieve both high coverage and improved homogeneity, the highest $\overline{SI}$ investigated (10,000 rpm) appears to be the most effective choice for the given material system and composition. However, at these high collision energies, potential degradation effects should be monitored, including, for instance, structural damage and strain in the NCM particles as well as elemental interdiffusion at the NCM|coating interface. Structural integrity was probed via scanning precession electron diffraction (SPED) in the TEM. In the dark-field TEM shown in Fig. 7a, the NCM particles appear dark, whereas the coating layer is brighter with distinct bright spots. In the region marked with a red box, a SPED phase map was recorded (Fig. 7b). Thereby, only the pristine layered phase and the coating layer were detected with no evidence of a rock salt-type phase in the NCM surface region, which would be indicative of NCM degradation. This observation was further confirmed by manually inspecting the diffraction patterns at the interface, which did not show any signs of degradation. Additional X-ray diffraction (XRD) measurements and Rietveld refinement likewise revealed no significant changes in lattice parameters or strain (Supplementary Fig. S12 and Table S3). The chemical stability at the interface was probed using STEM-EDX (Supplementary Fig. S13). The element map (Supplementary Fig. S13a) shows the CB-LIC matrix around the NCM particles. To check for elemental interdiffusion, a line scan across the interface was evaluated (Supplementary Fig. S13b). Due to the NCM surface being slightly slanted, the characteristic compositions of NCM and LIC overlap in a small interfacial region (Supplementary Fig. S13c). However, no evidence of elemental diffusion beyond this overlapping region of approximately 10–15 nm width was detected. Based on these observations, we assume that the coating process in the high-intensity mixer does not damage the NCM surface region. Rather, the small single-crystalline particles appear to tolerate the processing conditions. In addition, the softer LIC phase is expected to exert a dampening effect.

It is important to note, however, that this may differ when processing other material systems, such as larger polycrystalline CAM particles or sulfide SEs[73]. Larger particles exhibit a higher defect probability, typically, due to a lower particle strength than for aggregates, according to Rumpf[74], scales as $\sigma \propto \frac{1}{d^2}$, and for primary particles, according to Tavares and King[75], scales as $\sigma \propto \frac{1}{d^{1.2}}$, making fragmentation more likely at similar stress intensities. To corroborate this, we conducted a mixing experiment with larger polycrystalline NCM particles under the same high-stressing conditions applied in this study (10,000 rpm for 60 min). The results show that, although most particles retained their overall structure, a few fragmented particles were observed (Supplementary Fig. S14). Notably, these were consistently among the largest particles, which supports both our considerations. Moreover, applying a mill (e.g., planetary ball mill) for the composite

mixing can result in higher stress intensities and might also cause material degradation of the single crystal NCM[71,76].

In these cases, DEM simulations could provide guidance for process optimization, indicating that lower stress intensities, which might be necessary to prevent chemical or structural degradation during processing, can be compensated for by longer mixing times to achieve similar coverages. For example, in the case of coating at 1000 rpm, achieving a target coverage of approximately 97.5% would require extending the mixing time by 200%, i.e., to a total of ~3 h. When combined with further degradation studies, DEM simulations can help identify suitable process parameters for different material systems and mechanofusion processes.

## Electrochemical performance of mixed conducting matrix coatings

Based on the investigations discussed above, a maximum stress intensity and maximum stress number, i.e., a rotational speed of 10,000 rpm and a mixing time of 60 min, were selected as fixed process parameters to further study systematically the effect of carbon content in the matrix on the electrochemical performance of the resulting matrix coating. In the following, we refer to the CB-LIC premix as the matrix, and to the NCM–LIC–CB mixture, i.e., the coated NCM particles, as the composite.

Effective charge transport requires fast electronic and ionic pathways between the CAM particles and the current collectors. A matrix coating solely composed of low electron-conducting SE, i.e., a composition of 80:20:0, is expected to limit the electronic transport. In this case, electrons can only be conducted via NCM-NCM point contacts after deformation of the SE coating layer. Thus, the idea is to add an electron-conductive additive to form a mixed conducting matrix. A similar mixture, composed of CB and LPSCl, was already investigated by Reisacher et al.[77] to identify the necessary content of CB to achieve an electronically percolating matrix. Similarly, we evaluated different CB-LIC ratios to identify the percolation threshold $p_c$ for our system.

To this end, we determined the partial electronic conductivities using direct current (DC) polarization in symmetric ion-blocking cells, both for the CB-LIC matrix as well as for the NCM–LIC–CB (80:20:$x$) composites with different compositions. As shown in Fig. 8a, the threshold for electronic percolation, $p_c$, in the matrix lies between 2 and 5 wt%. Below this range, the partial electronic conductivity of the matrix remains below 0.01 mS cm$^{-1}$, while at 5 wt%, it increases by four orders of magnitude to almost 200 mS/cm and further rises up to 1000 mS cm$^{-1}$ at 15 wt% CB. The cathode composites exhibit a similar trend for the electronic partial conductivity, which increases from approximately 0.03 mS cm$^{-1}$ at 0% CB to 1.8 mS cm$^{-1}$ at 15% CB in the matrix.

Since the mixed conducting matrix not only needs to provide electronic but also sufficient ionic conduction, we further measured the ionic partial conductivities of the different matrices and composites, respectively. Using DC polarization in symmetric electron-blocking cells, an opposite trend for the ionic conductivity as a function of the CB content is observed (Fig. 8a). With increasing CB content, the ionic conductivity of the matrix decreases from about 0.7 to 0.03 mS cm$^{-1}$. Similarly, for the composites, the ionic partial conductivity decreases as well. For the highest CB content, the minimum ionic partial conductivity of about 0.006 mS cm$^{-1}$ was measured for the respective composite. The highest ionic partial conductivity of about 0.04 mS cm$^{-1}$ was found for the composite with a CB content in the matrix of 1 wt%.

The differences between maximum and minimum ionic conductivity values are not as large as for the electronic conductivity. Furthermore, not all data points follow a strict monotonic trend with increasing CB content. We attribute this observation to the fact that

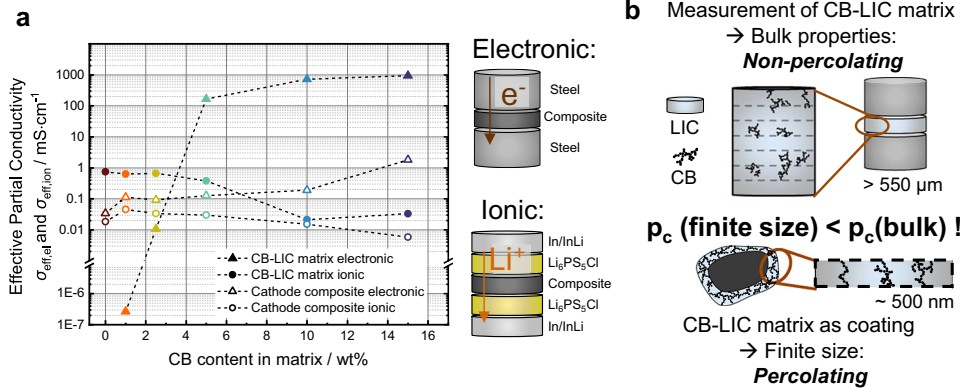

**Fig. 8 | Carbon black-dependent electronic and ionic transport properties and size-dependent percolation behavior in composite materials. a** Effective partial electronic and ionic conductivities of the matrix and composite as a function of CB content. **b** Schematic showing the impact of sample dimensions on measurable percolation behavior.

the CB is not equally homogeneously distributed in all samples, leading to different microstructures. In general, the differences in conductivities deviating from the trend are rather small, given the general level of uncertainty. Moreover, not only are the materials error-prone, but also the experimental conditions in the dry room might lead to irregularities.

At this point, we would like to highlight that the percolation behavior is not only dependent on the intrinsic material properties but also on the sample size of the considered volume used to determine the percolation threshold. During the measurement of the matrix, bulk properties are assessed, as a relatively large amount of powder is used to produce a thick pellet (Fig. 8b). The values obtained from these bulk-scale measurements are not necessarily representative of the situation when the matrix is applied as a thin coating. In this case, the coating is typically in the nanometer range, representing a finite-size system in which the percolation threshold $p_c$ can be reached earlier, i.e., at lower CB contents. Thus, the macroscopic bulk $p_c$ derived from symmetrical cell setups should rather be viewed as an upper limit for the CB content in the composite.

Additionally, partial conductivities primarily describe charge transport through the bulk composite and offer no insights into the charge transfer kinetics at the CAM/SE interface, particularly since the CAM is typically fully lithiated during such measurements[35]. Therefore, the partial conductivities obtained from symmetrical cells do not necessarily correlate fully with the half-cell performance, as recently also observed by Puls et al.[78].

This assumption is indeed reflected in the cycling performance of the half-cells depicted in Fig. 9a. Here, the cells with CB contents above the determined $p_c$ (80:20:1/2/3) show a significantly worse performance than cells with CB contents below $p_c$. While the attainable capacities at 0.1 C are still comparable, they rapidly decrease at 0.3 C and 1 C. For example, while providing an initial composite-specific capacity of -130 mAh g$^{-1}$ at 0.1 C, the 80:20:3 composition can only supply a discharge capacity of less than 10 mAh g$^{-1}$ at 1 C. Interestingly, the composite without any CB in the matrix, i.e., with a negligible electronic conductivity, exhibits a decent performance retaining approximately 85 mAh g$^{-1}$ at 1 C when being charged in constant current constant voltage (CCCV) mode. The electronic transfer across NCM-NCM point contacts seems to be sufficient at the tested stack pressure of about 80 MPa, confirming the previous results of Kim et al.[39]. In the here presented work, we could achieve further improvement in cycling performance when adding some CB into the matrix. Among the tested compositions, the 80:20:0.5 composite delivers the highest composite-specific discharge capacity of $q_{comp}=100$ mAh g$^{-1}$ at 1 C after charging in CCCV mode. In the

following, the cycling behavior depending on the cathode compositions shall be discussed in more detail.

All capacity values in Fig. 9a refer to the total mass of the composite, since this metric is closer to practically relevant performance indicators than CAM-specific capacities, which are shown together with the Coulombic efficiencies in Supplementary Fig. S15 (see also Supplementary Note 4). As discussed in our previous work[34], not all CAM particles are necessarily electrochemically active in SSB composite cathodes. Depending on the cathode composition and/or the mixing method, a substantial fraction of CAM can remain electronically disconnected and thus inactive, thereby lowering the overall static CAM utilization. Consequently, capacity values based on either the total mass of CAM or the total cathode mass reflect not the intrinsic active material properties but are strongly convoluted with the microstructure of the composite. Thereby, it is possible to differentiate between static and kinetic capacity losses[34]. Static capacity losses consider the issue of incomplete static CAM utilization which is non-negligible for the tested compositions as depicted in Fig. 9b. The highest CAM utilization of about 90% is reached for the highest CB content, while in the low CB (0–0.5%) containing composites only about 65% of the CAM particles are electronically connected. This is reasonable since CB in the matrix is expected to facilitate the formation of electronic connections between the NCM particles. Thus, when more CB is used in the matrix, the probability of electronic connection of a NCM particle increases. For the 80:20:3 composition this leads to the highest achievable capacity of 129 mAh g$^{-1}$ at low C-rates (see also Supplementary Figs. S15a and S16, and Table S4).

On the other hand, we observe that high CB contents are detrimental to the cell kinetics in this study. This is evident from the active specific capacities shown in Fig. 9b which are calculated by referring the measured charge only to the mass of active CAM. Hence, these capacity values describe the kinetics only of the statically active particles. For the 80:20:0.5 composition, an active specific capacity of 180 mAh g$^{-1}$ at 1 C is obtained while for the 80:20:3 composition, only about 10 mAh g$^{-1}$ are reached. The differential capacity curves depicted in Fig. 9c for discharge cycles at 0.3 C and 1 C confirm the differences in kinetic performance depending on the CB content. This is further supported by Supplementary Fig. S16 which shows that more charge is gained from the CV period for the samples with higher CB content, indicating available but kinetically limited capacity.

For an overall assessment, the active specific capacity values, representing a kinetic property, must be considered alongside CAM utilization, which reflects a static property[34]: A high CB content increases the CAM utilization by providing more electronic pathways, leading to higher capacities at low C-rates. However, it simultaneously

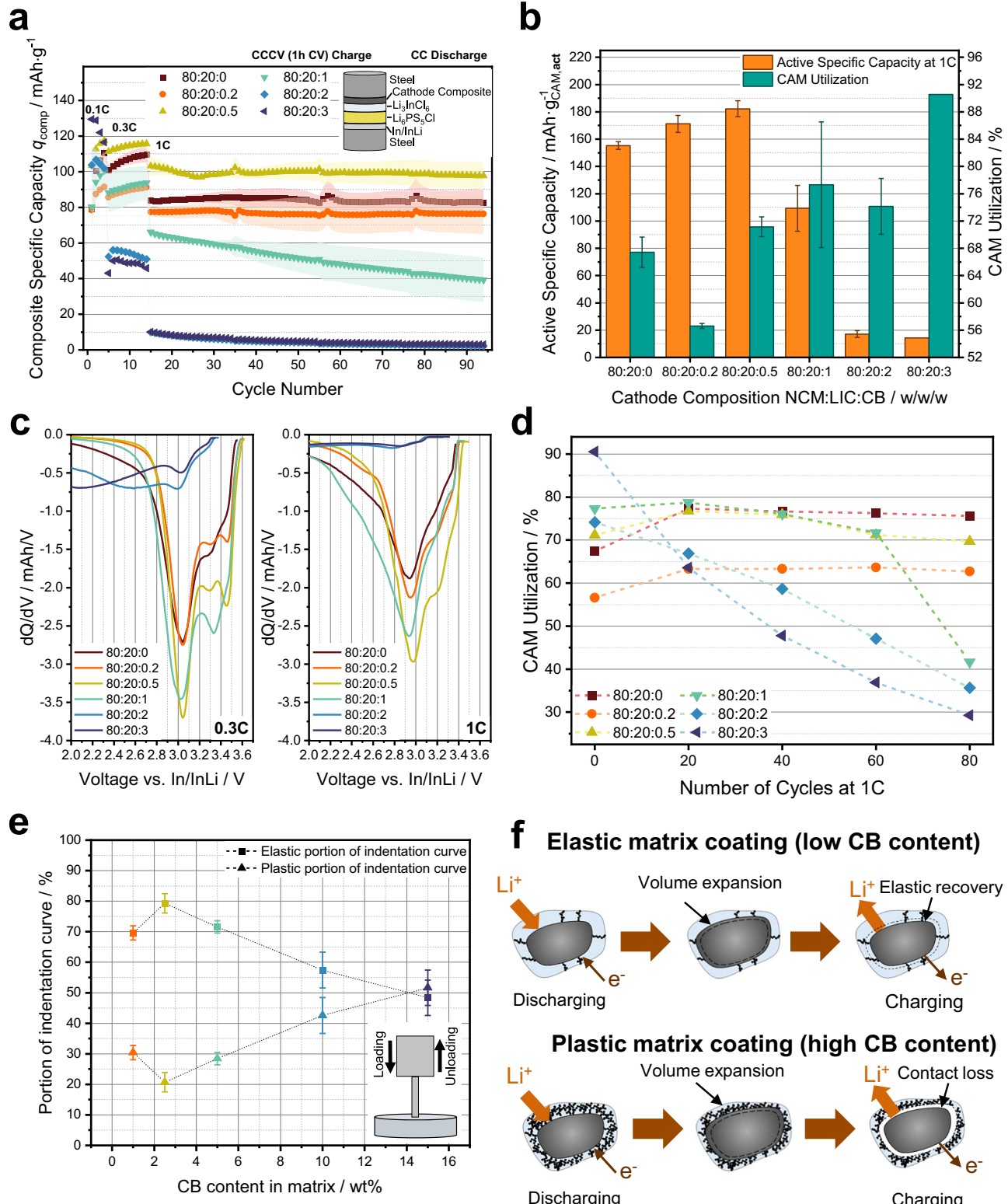

**Fig. 9 | Electrochemical performance of half-cells depending on the cathode composite composition ($1\,C \approx 200\,mA/g \approx 3\,mA/cm^2$). a** Cycling performance when cycled in CCCV mode at 25 °C under 80 MPa stack pressure. Error bars represent the standard deviation of at least $n = 2$ cells. **b** Comparison of active specific capacity and CAM utilization. Error bars represent the standard deviation of at least $n = 2$ cells. **c** Differential capacity plots at 0.3 C and 1 C. **d** Evolution of CAM utilization during cycling. **e** Elastic and plastic portions of force−indentation curves obtained from nanoindentation on CB-LIC pellets, which were prepared by compacting the premix at 380 MPa. Error bars represent the standard deviation of $n = 24$ indentations. **f** Schematic illustration of how the mechanical properties of the coating influence microstructure evolution. Elastic coatings enable elastic recovery, preserving interfacial contact, whereas plastic coatings lead to contact loss due to irreversible deformation.

obstructs the ionic pathways, thereby impairing the kinetic cell performance resulting in low capacities at higher C-rates. Conversely, a low CB content reduces the static CAM utilization, with many CAM particles remaining electronically disconnected and thus electrochemically inactive. At the same time, the remaining electrochemically active particles are found to be kinetically well connected, which is reflected in the high active specific capacities at 1 C (Fig. 9b) and lower overpotentials (Supplementary Fig. S16). The improved kinetic performance during charging and discharging is further underlined by the fact that these particles experience a significantly higher effective C-rate due to the low CAM utilization[34,79]. We do not observe a monotonic trend in capacity as a function of CB content, which we attribute to the fact that the CB distribution within the matrix coating is not yet fully optimized. Consequently, CB agglomeration may lead to reduced electronic connectivity, particularly at CB contents at or below the percolation threshold. This highlights the need to further optimize CB distribution in the matrix, which should be sufficient to ensure high CAM utilization, yet low enough to maintain fast transport kinetics and to prevent excessive electrochemical SE degradation.

In order to quantify the impact of chemo-mechanical degradation, i.e., contact loss[27,33], during cycling at 1 C in CCCV mode, the CAM utilization was determined every 20 cycles. Figure 9d indicates that contact loss, which results in loss of active mass, is more pronounced for the cells with high CB content. We hypothesize that the observed behavior stems from the mechanical properties of the matrix. With increasing CB content, the matrix is much more porous, exhibits reduced elasticity, and shows a greater tendency toward plastic deformation which may hinder its ability to accommodate the volume changes of NCM during continuous cycling. In contrast, a matrix with low CB content is more elastic and maintains contact with the NCM during cycling. For these composites the CAM utilization slightly increases and appears to stabilize during cycling, indicating microstructural changes which lead to new electronic connections.

Nanoindentation measurements on the CB-LIC premix (Supplementary Fig. S17), from which we calculated the elastic and plastic portions of the indentation response (Fig. 9e), corroborate this picture. While there is approximately 20% plastic portion at 2.5 wt% CB in the matrix, it rises to around 50% at 15 wt% CB, highlighting a substantial shift toward plasticity at higher CB concentrations. This behavior is accompanied by a substantial rise in porosity of the CB-LIC matrix with increasing CB content (Supplementary Fig. S17). Notably, at CB contents of 10% and 15% in the matrix, the porosity increases sharply from below 5% (at low CB levels) to approximately 25 and 30%, respectively. The pronounced increase in porosity is directly linked with the observed rise in plasticity and correlates with the sharp decline in CAM utilization during long-term cycling at high CB contents in the coating. As the matrix becomes mechanically less elastic, it loses its ability to accommodate microstructural changes, such as the volume expansion of the CAM, during cycling, leading to significant contact losses and degradation in electrochemical performance (Fig. 9f).

Overall, our results reveal a trade-off between CAM utilization, charge transport kinetics, and mechanical integrity, with a composition of 80:20:0.5 leading to the best performance among the compositions tested in this study. Thereby, no direct correlation between the partial conductivities and half-cell performance is found. The results indicate that the percolation threshold value determined in a macroscopic measurement is rather too high and leads to poor ionic conduction as well as unfavorable mechanical properties which results in a worsened half-cell performance. In general, the concept of partial conductivities appears to have limited meaning in the context of the mixed conducting matrix coating approach introduced in this study. Especially mechanical properties and porosity of the CB-LIC premix as well as the interfacial contact quality are not reflected in these measurements although they were shown to

play a significant role. A more relevant assessment can be made by distinguishing between static and kinetic capacity losses in half-cell performances during cycling.

## Discussion

Interfacial engineering remains a key challenge for SSBs, as electrochemical, chemical, and especially mechanical instabilities at interfaces strongly affect cycling stability and rate performance. Among various mitigation strategies, solution-based and mechanochemical coating routes have emerged as two complementary approaches to optimize SSB microstructures.

Solution-based routes may enable the formation of SEs around CAM particles in-situ. For the halide SE $Li_3InCl_6$, such coatings have been demonstrated for $LiCoO_2$ by Wang et al.[80] and Ma et al.[81] using aqueous precursor solutions, followed by (freeze-)drying to crystallize the coating phase. This one-step route is attractive for scalability and energy efficiency, since no separate SE synthesis is required. However, aqueous processing is limited to water-tolerant CAMs like $LiCoO_2$, as Ni-rich CAMs can degrade upon water exposure[82–84]. To date, water-based synthesis has only been reported for $Li_3InCl_6$[85], while other halide SEs and especially sulfide SEs are chemically unstable in water or alcohols[55,86,87]. Consequently, toxic or highly flammable solvents such as xylene, toluene, or acetonitrile are required[88–91]. This introduces safety risks and higher processing costs due to significant energy demand for solvent evaporation and recovery.

Mechanochemical coatings, as presented in this study, offer a solvent-free alternative, avoiding interactions of the solvent with the CAM. High-shear mixing deposits pre-synthesized SEs directly onto CAMs, ensuring intimate contact and large interfacial area without SE pre-comminution or drying steps[60,61,63,65]. Pre-synthesized SE powders (e.g., LIC or sulfides) are mechanically attached to CAM particles without any intensive SE particle pre-comminution or drying step, eliminating solvent use and associated environmental and safety concerns. Although SE pre-synthesis is still needed, advances in large-scale mechanochemical synthesis have reduced its energy demand[15,92], potentially enabling a fully solvent-free SSB manufacturing chain[76].

As shown in this work, coating formation and homogeneity in dry mechanofusion processes are governed by the interplay between various process parameters, requiring precise process optimization. Excessive collision intensities can induce mechanical degradation of the CAM or SE[76], while insufficient stress intensity and/or number leads to incomplete coatings. The risk of particle defects or breakage increases for large or polycrystalline CAMs due to their higher defect probability. Conversely, small, single-crystalline CAM particles exhibit superior mechanical stability, which explains why no measurable degradation was observed even at the highest stress intensities (10,000 rpm, 60 min) within the mixer used in this study. Further performance improvement is expected by optimizing the homogeneous dispersion of CB in the matrix[93–95]. In particular, preventing agglomeration of the conductive additive should enhance CAM utilization while preserving fast interfacial charge transfer and transport throughout the composite cathode. Since both mechanochemical and solution-based coating strategies have distinct advantages and limitations, their applicability will depend on the specific material system, thereby motivating further process optimization and degradation studies.

In summary, we demonstrate that high-intensity mixing leading to mechanofusion is a promising, well-scalable dry-processing approach to fabricate tailored microstructures. We exploit the plastic deformability of $Li_3InCl_6$ as a model SE, to create nanometer-thin coatings on the one hand, while also being suitable for building up well-defined thicker mixed conducting matrix coatings around single-crystalline NCM on the other. The process can be broadly applied across different types of SEs. Unlike simple mixing of loose particles and arbitrary (re-)ordering, the mechanofusion process offers a bottom-up pathway to

engineer cathode building blocks with good interfacial connectivity, eliminating the need for manual mixing or multiple post-processing steps. Through systematic variation of process conditions and compositions, supported by DEM simulations and advanced analytics, we identified key parameters controlling the coating morphology, quantified by XPS-derived coverage values, and electrochemical performance. Notably, cathodes with a composition of 80:20:0.5 (NCM:LIC:CB, w/w/w) exhibited stable cycling at 1 C and a specific capacity of $q_{comp} = 100$ mAh g$^{-1}$, calculated with respect to the total mass of the composite cathode. We expect considerably higher capacities once SEs with much higher conductivity in the order of 10 mS cm$^{-1}$ will be used. Our findings indicate a clear trade-off between CAM utilization, cell kinetics at higher CB contents, and the mechanical behavior of the matrix coating, underlining the importance of optimizing the composite mixing process. Overall, this work presents a promising and scalable strategy for advancing the manufacturing of high-performance SSB cathodes. While also discussing corresponding challenges of this approach, it motivates further exploration of engineered particle architectures in solid-state energy storage systems.

## Methods

### Materials
Single-crystalline NCM82 (LiNi$_{0.82}$Mn$_{0.07}$Co$_{0.11}$O$_2$ from MSE Supplies, Tucson, AZ, USA), CB (Super C65 from Imerys, France), and in-house synthesized Li$_3$InCl$_6$ (LIC) were used. For the latter one, a water-mediated synthesis was carried out in a dried room with a dew point of $-60$ °C. The starting components LiCl (99%, Fisher Scientific, Waltham, MA, USA) and InCl$_3$ (99.995%, Fisher Scientific, Waltham, MA, USA) with a mass ratio of 0.365:0.635 (LiCl:InCl$_3$) were mixed in deionized water at 40 °C for 2 h using a stir fish. Then, the mixture was first dried at 80 °C with the vessel open for 24 h and afterwards dried under vacuum for 4 h at 200 °C. From each batch, 32 g of dried in-house synthesized LIC was obtained, and this water-mediated synthesis route could be easily scaled up by employing larger vessels. After drying, the LIC was comminuted using a planetary ball mill (Pulverisette 7, Fritsch GmbH, Idar-Oberstein, Germany) at 600 rpm for 4 min with 5 mm ZrO$_2$ milling beads, using a 50% milling bead filling ratio.

### High-intensity mixing
The building blocks were produced using a Picoline high-intensity mixer equipped with a Nobilta attachment (Hosokawa Alpine AG). Mixing was performed in a dry room maintained at a dew point of $-60$ °C of the supply air. To mitigate the considerable heat generated during mixing, continuous cooling was applied. The material volume that was filled in was consistently set to 4.91 mL based on the calculated pure mixed density $\rho_{mix}$ across all tests to ensure comparability of stressing conditions (cf. Table S1). In the CB-containing samples, CB and LIC were premixed at 5000 rpm for 10 min to pre-structure the matrix coating. NCM was subsequently added, and mixing was continued for up to 60 min. Sampling during the rotational speed variation tests was performed at 5, 10, 30, and 60 min. All mixtures were stored in sealed bags in an argon atmosphere within the dry room prior to further analysis.

### Power data evaluation
The power data $P$ of the high-intensity mixer was continuously recorded at 5 s intervals throughout the entire mixing process. For the evaluation of the experimental specific energy input $E_{m, exp}$, only the 60 min mixing phase of all three components was considered. To smooth the power signal, a Savitzky-Golay filter with a span of 50 data points was applied. The idle power $P_0$ corresponding to each rotational speed was subsequently subtracted from the smoothed data. The resulting net power $\Delta P$ was then normalized to the mass of the composite $m_{total}$ used in each experiment (cf. Table S1). Finally, the experimentally derived specific energy input $E_{m, exp}$ was calculated with the corresponding mixing time $t_{mix}$ according to Eq. 3.

$$E_{m, exp} = \frac{\sum_0^{t_{mix}} \Delta P \cdot \Delta t}{m_{total}} \tag{3}$$

### Particle size analysis
To determine the particle size of the building blocks, three SEM images were analyzed using an in-house artificial intelligence tool (particleO-S.AI). Details are found in Supplementary Note 1.

### DEM simulations
Coarse-grained DEM simulations were conducted to capture the macroscopic processing conditions inside the high-intensity mixing device using Rocky version 2024 R1. To reduce computational cost and enable simulations within a practical timeframe, the particle size was coarse-grained to 300 μm, approximately 100 times larger than the actual experimental particle size.

Calibration experiments and simulations were performed to reproduce the particle dynamics and material behavior. First, compaction experiments were carried out using a ZWICK materials testing machine (ZwickRoell GmbH & Co. KG, Ulm, Germany). In these tests, 1 g of the 80:20:3 composite (processed at different rotational speeds for 60 min) was compacted up to 1000 N, and the corresponding force–displacement curves were recorded. In the DEM simulations, an equivalent 1 g of coarse-grained particles was placed inside a virtual compaction cylinder and compressed until a force of 1000 N and the same average displacement as in the experiments was reached. By adjusting the yield ratio, Young's modulus, and friction parameters, the mechanical response of the simulated particles was calibrated to match the experimental results (cf. Supplementary Fig. S10a).

To further capture the particle flow behavior, the dynamic angle of repose of the 80:20:3 composition was determined using a Granu-Drum (Granutools, Awans, Belgium) at a rotational speed of 20 rpm and 51.2 g of powder, during which 60 images were recorded and averaged. In the simulations, a corresponding slice of the drum was modeled and filled with coarse-grained particles rotated at the same speed. As only 15% of the drum length was simulated, only 7.68 g of coarse-grains were used in the simulation to save computation time. The static and dynamic friction coefficients were then iteratively adjusted to reproduce the experimentally observed angle of repose (cf. Supplementary Fig. S10b). The set of calibrated parameters used in the simulation is provided in Supplementary Table S2.

The coarse-grained particles were assumed to possess an internal porosity of 0.39. Based on an experimentally determined bulk density of 1.45 g cm$^{-3}$ and a theoretical mixed density of 3.94 g cm$^{-3}$ for the 80:20:3 composition, a corresponding coarse-grain density of 2.417 g cm$^{-3}$ was calculated (cf. Supplementary Fig. S10c). To validate this value, the filling volume determined from the GranuDrum experiments was compared with the simulated filling volume, showing a high agreement (cf. Supplementary Fig. S10d).

To replicate the 80:20:3 mixing experiment at different rotational speeds, the mixing chamber of the high-intensity mixer was virtually filled with 3.88 g of material (one-fifth of the material) via two particle inlets over a time span of 0 s to 0.1 s., and only one-fifth of the mixer volume was simulated, following the approach by Frankenberg et al.[43]. Following the filling phase, the rotor was set to rotate at varying speeds for 0.4 s until the steady state was reached. The mechanical power acting on the rotor was computed and compared to the experimentally recorded net power, enabling validation of the calibration of the simulation parameters from the compaction and angle of repose experiments. The elasto-plastic Thornton–Ning contact model was employed to account for both the plasticity and elasticity of the particle species.

In addition to the overall power acting on the rotor, particle-based (i) mean dissipated power $\overline{P_i}^\alpha$ was extracted during the output

intervals of $\Delta t_{\text{out}} = 0.005$ s and evaluated following the methodology described by Frankenberg et al.[43]. Therefore, the power values, which are a sum of the work done at one particle $\sum_{c=1}^{N_{c,i}} W_c^{\text{diss}}$ during the output interval, was divided by its collision frequency $c_i = \frac{\Delta t_{\text{out}}}{N_{c,i}}$ and the mass of one coarse-grain particle $m_{\text{particle, CG}}$ to obtain particle-based stress intensities. These stress intensities for each output interval were stored $N_{c,i}$ times to consider that particles undergo $N_{c,i}$ collisions during the output interval. From this data, stress intensity $SI$–stress frequency relationships were derived, and the overall mean stress intensity $\overline{SI}_{\text{CG}}$ of this distribution, as well as the coarse-grain stress frequency $SF_{\text{CG}}$ representing the number of all coarse-grain collision events of all particles per second, were derived. The stress number per particle $\overline{SN}_p$, corresponding to the total number of stress events per particle over the process time, was then calculated by integrating the stress frequency over time[43].

$$\overline{SI}_{\text{CG,i}}^{\text{diss}} = \frac{P_i^{\text{diss}}}{c_i \cdot m_{\text{particle, CG}}} = \frac{\sum_{c=1}^{N_{c,i}} W_c^{\text{diss}}}{\Delta t_{\text{out}}} \cdot \frac{\Delta t_{\text{out}}}{N_{c,i}} \cdot \frac{1}{m_{\text{particle, CG}}} \quad (4)$$

$$\overline{SN}_p = SF_{\text{CG}} \cdot t_{\text{mix}} \cdot \frac{m_{\text{particle, CG}}}{m_{\text{total}}} \quad (5)$$

## XRD analysis

XRD patterns were recorded using a diffractometer equipped with a Cu K$\alpha$ monochromatic source ($\lambda = 0.154$ nm; Empyrean, Malvern Panalytical, Kassel, Germany). Measurements were performed over an angular range of 5° to 120° 2$\theta$ with a step size of 0.053° 2$\theta$. Samples were prepared in an argon atmosphere and sealed with Kapton foil to prevent air exposure during analysis. Background subtraction was applied to all spectra.

Additional XRD measurements to determine the influence of the mixing process on micro-strain were performed using a diffractometer equipped with a Mo K$_\alpha$ source (= 0.071 nm; Empyrean, Malvern Panalytical, Kassel, Germany). Measurements were performed over an angular range of 5° to 40° with a step size of 0.053° using capillaries with 0.6 mm diameter in transmission mode and a divergent beam path. Samples were prepared in an argon atmosphere.

## Nanoindentation

Nanoindentation measurements were carried out on pellets made from CB-LIC premixes with varying compositions. The pellets were prepared by compressing 0.69 g of premix at 380 MPa for 3 min using a laboratory press (LaboPress P200S, Vogt Labormaschinen, Germany). Each pellet was then affixed to a glass slide using epoxy resin. To protect the sample from air exposure during transfer, a plastic collar was mounted around the pellet, and a glass cover slip was sealed on top with high-viscosity paste.

Measurements were conducted using a TriboIndenter system (Hysitron Inc., Minneapolis, MN). The sample chamber was purged with argon prior to testing to prevent degradation of the pellet. A 100 μm diameter flat punch indenter was used in displacement-controlled mode. Indentations were performed to a depth of 10 μm at a loading rate of 500 nm s$^{-1}$. For each pellet, 25 indentations were made, and the average force–displacement curve was determined. From the curves, both the elastic $W_{\text{elastic}}$ and plastic $W_{\text{plastic}}$ work portions of the total indentation work $W_{\text{total}}$ were quantified.

$$W_{\text{total}} = \int_0^{s_{\text{max}}} F_{\text{loading}}(s)\,ds \quad (6)$$

$$W_{\text{elastic}} = \int_{s_0}^{s_{\text{max}}} F_{\text{unloading}}(s)\,ds \quad (7)$$

$$W_{\text{plastic}} = W_{\text{total}} - W_{\text{elastic}} \quad (8)$$

Here, $s_{\text{max}}$ denotes the maximum indentation depth at the highest applied load, while $s_0$ corresponds to the residual indentation depth after unloading (i.e., the intersection with the $x$-axis). $F_{\text{loading}}$ and $F_{\text{unloading}}$ represent the measured force during the loading and unloading phases, respectively.

## Determination of porosity

The porosity of the composites and pure materials was measured in a slightly compacted state using a ZWICK materials testing machine (ZwickRoell GmbH & Co. KG, Ulm, Germany). For each sample, three measurements were performed by filling 1 g of composite powder into the measuring cylinder of the instrument. For the individual raw materials, the following masses were used: 2 g for LIC, 0.25 g for CB, and 1 g for NCM. Compaction was performed up to a force of 395 N (3.45 MPa), and the displacement and applied force were recorded. The bulk density $\rho_{\text{bulk}}$ of the composite was determined geometrically in the compacted state, and porosity $\varepsilon$ was calculated based on the mixed theoretical solids density $\rho_{\text{mix}}$ derived from the known raw material densities, which are $\rho_{\text{NCM}} = 4.75$ g cm$^{-3}$, $\rho_{\text{LIC}} = 2.59$ g cm$^{-3}$ and $\rho_{\text{CB}} = 1.96$ g cm$^{-3}$ (cf. Table S1).

$$\rho_{\text{mix}} = vol.\%_{\text{NCM}} \cdot \rho_{\text{NCM}} + vol.\%_{\text{LIC}} \cdot \rho_{\text{LIC}} + vol.\%_{\text{CB}} \cdot \rho_{\text{CB}} \quad (9)$$

$$\varepsilon = 1 - \frac{\rho_{\text{bulk}}}{\rho_{\text{mix}}} \quad (10)$$

The porosity of the CB-LIC pellets used for nanoindentation measurements was determined after compaction at 380 MPa.

## Scanning electron microscopy (SEM)

The sample preparation was performed in an argon-filled glovebox with oxygen residues of $p(\text{O}_2)/p < 1.0$ ppm and water residues of $p(\text{H}_2\text{O})/p < 1.0$ ppm. The particles were imaged using a field-emission SEM *GeminiSEM 560* system (Carl Zeiss Microscopy GmbH, Jena, Germany) with 2 kV acceleration voltage and 3 mm working distance. EDX analysis was carried out with the *AzTec* EDX system (Oxford Instruments, United Kingdom). EDX mapping was performed with an acceleration voltage of 10 kV at a working distance of 8.5 mm using the *Ultim Max* detector. Prior to imaging, the particles were sputtered with a 4 nm thin layer of platinum using a LEICA EM ACE600 coater (Leica Microsystems GmbH, Wetzlar, Germany).

## (Scanning) transmission electron microscopy ((S)TEM)

The sample preparation of the powder sample was performed in an argon-filled glovebox with oxygen residues of $p(\text{O}_2)/p < 1.0$ ppm and water residues of $p(\text{H}_2\text{O})/p < 1.0$ ppm. Here, the particles were spread onto carbon film-coated Cu-mesh TEM grids. Moreover, a TEM lamella was prepared by focused ion beam (FIB) milling using a Helios$^{\text{TM}}$ 5 Hydra CX DualBeam System (Thermo Fisher Scientific Inc.). The sample was transferred in an argon atmosphere from the glovebox to the FIB system and back using the CleanConnect inert gas transfer system. The TEM grids were then loaded in the glovebox into an inert gas/vacuum transfer TEM holder (from Mel-Build) to avoid reaction of the samples with humidity in the air and transferred to a pumping stand. Here, the argon inside the holder was removed before inserting the holder into the TEM.

High-resolution as well as bright-field and dark-field TEM images were recorded on both a JEOL JEM 3010 (300 kV) and a double Cs-corrected JEOL JEM 2200 FS (200 kV). Both microscopes are equipped with a TVIPS TemCam XF416FS camera, which was used for image acquisition.

The SPED phase map was measured on the JEOL JEM 3010 equipped with a NanoMEGAS P2000 ASTAR system for scanning and precessing the beam (precession angle 0.6°). The four-dimensional data set was recorded with the TVIPS TemCam XF416FS camera as a video, which was later converted into a rectangular data set (.bloc format) using an in-house written Python code and evaluated using the NanoMEGAS software suite DiffGen 2, Index 2, and MapViewer 2. For phase matching, the crystallographic files (.cif) for NCM82, NiO, and LIC taken from the Inorganic Crystal Structure Database were used as input.

EDX data was recorded on the JEOL JEM 2200 FS using a Bruker Nano XFlash Detector 5060 in scanning TEM mode with a semi-convergence angle of 15.07 mrad. The map data was evaluated using the Esprit 2.3 software.

## X-ray photoelectron spectroscopy (XPS)

XPS analysis was performed using a PHI Versa Probe IV system (Physical Electronics Inc., Chanhassen, MN, USA). The powders were filled into Teflon crucibles (inner diameter: 3 mm), pressed to achieve a flat surface, and attached to the sample holder using nonconductive adhesive tape. The sample preparation was performed in an argon-filled glovebox with oxygen residues of $p(O_2)/p < 1.0$ ppm and water residues of $p(H_2O)/p < 1.0$ ppm. All samples were transferred to the XPS machine in an air-tight transfer vessel. Monochromatic Al-$K_\alpha$ radiation (1486.6 eV) was applied for XPS analysis. The X-ray source was operated at a power of 50 W and a voltage of 15 kV, having a beam diameter of 200 μm. A pass energy of 55 eV, a step size of 0.2 eV, a step time of 25 ms, and 20 sweeps were used. The samples were charge-neutralized during measurements. The XPS data were evaluated using CasaXPS (Casa Software Ltd., Teignmouth, UK). The spectra were calibrated in relation to the signal of adventitious carbon C 1$s$ at 284.8 eV. For each detail spectrum, a region with the same energy boundaries was defined, and the respective intensities $I_{element}$ (in CPS) were used to calculate the coverage values $\gamma$[66]:

$$\gamma = \frac{I_{Cl} + I_{In}}{I_{Cl} + I_{In} + I_{Ni} + I_{Mn}} \qquad (11)$$

Equation 11 does not contain the C 1$s$ signal intensity because this would include signal intensity stemming from adventitious carbon, which cannot be distinguished from the CB contribution. Moreover, the overall signal intensity, including C 1$s$, depends on several experimental parameters and shows noticeable variations between nominally similar samples. Thus, even the corrected C 1$s$ intensity, obtained by subtracting the C 1$s$ intensity of a CB-free reference sample, was not used in the coverage calculation. XPS-derived coverage values close to 1 reflect that a covering, >10 nm thick, coating layer was achieved. The values contain no information on the morphology and exact thickness distribution. The coverage values contain averaged information over more than 1000 particles.

## Cell building and electrochemical evaluation

All electrochemical cells were built in an argon-filled glovebox with oxygen residues of $p(O_2)/p < 1.0$ ppm and water residues of $p(H_2O)/p < 1.0$ ppm. The cell components were filled in an in-house-built cell casing containing a PEEK die with an inner diameter of 10 mm[34]. The electrochemical measurements were carried out in a controlled environment at $T = 25$ °C. During all electrochemical measurements, the cells were fixed in a steel frame while applying a pressure of approximately 80 MPa (10 Nm torque).

For the determination of the effective partial electronic conductivity, symmetric ion-blocking cells were used with stainless steel rods serving as contacts. 120 mg of the matrix and composite powders were compacted with an uniaxial pressure of 380 MPa for 3 min at room temperature.

For the determination of the effective partial ionic conductivity, symmetric electron-blocking cells were used. For this purpose, 240 mg of composite was hand pressed, and then 60 mg of Li$_6$PS$_5$Cl (LPSCl, Argyrodite-CMP5 from Posco JK Solid Solution, South Korea) was added to each side. The three layers were compacted with a uniaxial pressure of 380 MPa for 3 min at room temperature. Then, indium foil (99.999%, 100 μm thickness, 9 mm diameter, Chempur, Germany) and lithium foil (100 μm thickness, 4 mm diameter, China Energy Lithium) were pressed together on both sides to serve as lithium reservoirs.

For the experimental measurement of effective partial conductivities, chronoamperometry was applied until a stable steady-state current was reached at every voltage step. The voltage-current tuples were fitted linearly, and the resistance and the corresponding conductivity were evaluated according to Ohm´s law. For the symmetric electron-blocking cells, an additional reference measurement using Li-In|LPSCl|In-Li was carried out. The associated resistance was subtracted from the total resistance before calculating the ionic partial conductivity.

For the half-cell testing in a in-house made cell casing[96], the following setups were used: First, 40 mg of LiPSCl powder (Argyrodite-CMP5 from Posco JK Solid Solution, South Korea) was evenly filled in a PEEK cylinder (10 mm diameter) and pressed by hand. Subsequently, a layer of 40 mg LIC was added and also hand pressed, completing the bilayer separator[97]. Then, approximately 15 mg of the cathode composite was evenly distributed on the LIC layer. On the LPSCl layer of the bilayer separator, an indium foil (99.999%, 100 μm thickness, 9 mm diameter, Chempur, Germany) and a lithium foil (100 μm thickness, 6 mm diameter, China Energy Lithium) were placed to form the In/InLi anode. Afterwards, the cell was uniaxially pressed with approximately 380 MPa for 3 min while being isolated.

The cells were cycled in a cell voltage range between 2.0 and 3.7 V vs. In/InLi at room temperature using a BCS-805 Battery Cycling System (Bio-Logic, Seyssinet-Pariset, France) and a MACCOR electrochemical workstation. Charging was done in CCCV mode by holding the cutoff potential for 1 h. The C-rates were calculated based on a CAM-specific capacity of 200 mAh/g$_{NCM}$, taking into account the nominal amount of NCM in the cathode (1 C ≈ 200 mA/g ≈ 3 mA/cm²).

The CAM utilization was determined similarly to as described in a previous publication[34]: Each SSB cell was charged with 0.1 C to 3.1 V vs. In/InLi, held at that potential for 3 h, followed by 3 h of relaxation to determine the potential $V_1$. Then, the cell was charged with 0.05 C up to 3.4 V, followed by 5 h of relaxation to get the potential $V_2$. The relaxed OCP values are given in the Supplementary Source Data File. As a quasi-OCP (titration) curve of the CAM, the capacity-voltage curve of an LE cell at 0.02 C was taken as a reference (cf. Supplementary Fig. S18) for which the data file is openly available in Zenodo at https://doi.org/10.5281/zenodo.14065128, reference number 14065128. For the LIB reference cell, it was assumed that it possesses a comparable voltage-capacity curve to the SSBs. The validity of this approximation is discussed in Supplementary Fig. S19.

The liquid cell data was used to assign a reference specific capacity value $q_{ref,LIB}$ to the potential range determined by $V_1$ and $V_2$. This value was compared with the measured capacity $q_{measured}$ of the solid-state cell, which can be extracted with the analysis software. The CAM utilization was then calculated with Eq. 11:

$$\text{CAM Utilization} = \frac{q_{measured}}{q_{ref, LIB}} \qquad (12)$$

The active specific capacities are calculated by dividing the measured capacities by the CAM utilization, i.e., the charge is referred only to the actual electrochemically active mass.

## Data availability

The source data used for all figures in this study are provided in the Source data File, which has been deposited, together with additional data, in Zenodo at https://doi.org/10.5281/zenodo.18493074.

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

## Acknowledgements

M.K., F.F., J.J., and A.K. acknowledge financial support by Deutsche Forschungsgemeinschaft (DFG, German Research Foundation) through the priority program 2289 (project 462470125) and by BMFTR through the projects 03XP0430A, 03XP0430C (FestBatt Cluster of Competence, FB2-Thio) and 03XP0590D (FoFeBat). Language and grammar of the manuscript have partially been improved with the help of *DeepL Write* and *ChatGPT4.0* (by OpenAI).

## Author contributions

M.K. and F.F. contributed equally to this work. M.K., F.F., A.K., and J.J. conceived the project. M.K. and F.F. designed and coordinated the experiments. F.F. prepared the mixtures with the help of N.L., measured the porosities, carried out DEM simulations of the mixing process and analyzed the data. M.K. carried out the electrochemical investigations with help of A.L., measured and analyzed the EDX-SEM and XPS data. T.D. carried out the STEM-EDX and TEM investigations. D.W. analyzed the XRD data. M.K. and F.F. wrote the first version of the manuscript which was edited by all authors.

## Funding

## Competing interests

The authors declare no competing interests.
