## [Transparent Peer Review file · Nature Communications]

Mechanofusion-derived cathode composite microstructures with scalable mixed conducting matrix coatings for solid-state batteries

Corresponding Author: Professor Juergen Janek

Version 0:

Reviewer comments:

Reviewer #1

(Remarks to the Author)

Please find the attached PDF document.

Reviewer #2

(Remarks to the Author)

The manuscript describes an interesting study that merits publication in Nature Communications. However, the level of analysis is lacking, especially for such a high-quality journal. Additionally, some assumptions are made that are not identified and that require justification. Therefore, I recommend major changes and further experiments prior to publication. Detailed comments follow.

L166

"Grain boundaries existing within agglomerates of the bare NCM disappear completely (Supplementary 167 Figure S2)." I cannot see any differences in grain boundaries in the images shown. Figure S2 is nice, but Figure 2a is too small to see these features. Larger versions should be shown (e.g. in the supplemental section), so that readers can see what is going on.

Cross section images should be shown to show the thickness of the coatings on the NCM particles. Moreover, it is known that the mechanofusion process can cause strain and structural damage to the surface of materials, particularly NCM. It has also been previously reported that the surface of cathode materials may become amorphous during mechanofusion processing. For this reason, TEM should be used to examine the surface of FIB cross sections, so that the cathode surface can be mapped more carefully, with electron diffraction used to detect structural changes. Moreover, the cathode/LIC interface should be examined closely (e.g. to see if any elements from the cathode become incorporated into the LIC).

A detailed analysis of the XRD diffraction patterns is needed. Especially Rietveld refinement should be performed on all sample XRD patterns (i.e. all compositions and all processing conditions) to detect any changes in lattice constants and cation mixing extent in the cathode after processing. Further analysis should be performed to derive values of the lattice strain and grain size (which is related to the defect concentration). This is highly important information for all samples, since the electrochemical performance of NCM is very sensitive to strain or the accumulation of defects. The XRD patterns shown in Figure S5 are also much too small. This figure should be made much larger, so that readers can see any changes in XRD patterns (e.g. expanding this figure to the size of an entire page would be nice).

L369

"This assumption is indeed reflected in the cycling performance of the full cells depicted in Fig. 8a."

All cells constructed in the manuscript cycled versus a InLi counter/reference electrode should correctly be referred to as "half-cells". (please see, for instance, <https://doi.org/10.1002/bte2.20220052> for a good discussion of the distinction between solid-state full cells and half-cells).

L385

"As discussed in our previous work (40), this can be misleading,"

Indeed. However, not showing the plot in units of mAh/(g NCM) can be equally misleading. This is because it is very possible that the processing conditions could cause a reduction in the capacity of NCM through defects or induced strain, which is not considered here. It needs to be seen if processing conditions, reduce the capacity of NCM. Therefore, the plot should be shown both ways.

L388

"Static capacity losses consider the issue of incomplete CAM utilization which is substantial for the tested compositions as depicted 389 in Fig. 8b."

The determination of CAM utilization and active specific capacity should be fully explained in the experimental section or in supplemental information. Currently, the reader is referred to Reference 40 for this explanation, where Reference 40 itself refers the reader to another reference. Therefore, no reference contains the complete description and the reader must go two references deep to find out what is going on. It is important to clearly explain this procedure, since the data shown in Fig. 8b assumes that the mechanofusion processed CAM has an identical voltage curve to the pristine CAM. This assumption needs to be communicated to the reader. Furthermore, it needs to be justified that this assumption is true, even though previous studies have shown that mechanofusion processing of NCM can cause major changes to the voltage curve (e.g. see <http://dx.doi.org/10.1149/2.0681913jes>).

Electrochemical characterization is insufficient. Full voltage curves and full differential capacity curves should be shown at the first cycle and subsequent selected cycles. A table should be provided with the following information for each composition: first reversible capacity, initial coulombic efficiency, and capacity fade/90 cycles. Plots of coulombic efficiency vs. cycle number should be shown. These properties should be discussed in relation to composition.

The "known raw material densities" of each component used to calculate the pure mixing density should be stated. It should also be stated clearly if the "known raw material densities" are particle densities or bulk densities and how these densities were determined.

The actual material amounts (in grams) of each component used in the high intensity mixing step should be stated (e.g. in a table in the supplemental section).

L635

"First, 40 mg of LiPSCI (Argyrodite-635 CMP5 from Posco JK Solid Solution, South Korea) was pressed by hand."

This statement is vague. Maybe it means that the powder was put into the cell in an even layer and then pressed by hand?

L638

"On the LPSCI side of the separator"

Please clarify what is meant by "separator".

L643

"The C-rates were calculated based on a specific capacity of 200 mAh/g"

This is vague. The capacity was based on 200 mAh per gram of what material?

Reviewer #3

(Remarks to the Author)

The manuscript presents a compelling mechanofusion approach for creating a mixed conducting matrix on NCM cathodes for solid-state batteries, demonstrating promising electrochemical results. The use of discrete element method (DEM) simulations is a significant strength. However, some clarification and additional data are required to fully support the conclusions.

1. NCM Particle Integrity and Structural Stability: The reviewer observed cracks in NCM particles after mechanofusion (Figure 1d) and is concerned about the structural stability of the cathode under the intense processing conditions (10,000 rpm/60min). Please provide a thorough explanation of how the NCM particles maintain their integrity, supported by a quantitative assessment of particle damage (e.g., wider-field SEM images) across different processing stages. The discussion should also address the mechanical properties of NCM that enable this stability and the potential impact on electrochemical performance.
2. Comparison to Solution-Based Methods: The manuscript should explicitly compare the high-energy mechanofusion process to more common solution-based coating methods (e.g., Sun et al., Nano Energy 76 (2020) 105015). This comparison should cover: (1) Processing complexity and simplicity: How do the two approaches differ in terms of required equipment and steps? (2) Scalability and commercial viability: Is mechanofusion a practical method for large-scale production? (3) Resultant coating quality and performance: How do the final coatings and electrochemical performance compare? The authors need to justify why the energy-intensive mechanofusion process is a preferable method despite its apparent drawbacks.
3. Side Reactions and Interfacial Stability: Given that prolonged high-energy processing can exacerbate side reactions between LIC and NCM, leading to the formation of ion-electron insulating byproducts at the cathode-electrolyte interface (CEI), the authors must explicitly address how the impact of these reactions is minimized or managed.
4. Data Analysis Discrepancy (Coating Thickness): The linear relationships between coating mass and thickness shown in Figures 2b, 2c, and S3 are inconsistent with the physical properties of the materials. Since LIC and carbon black (CB) have different densities, the slope of the linear regression should change when CB is added, but the manuscript shows consistent slopes. The authors need to re-evaluate this data, account for the density differences, and provide a clear explanation for how they derived accurate and credible results.
5. Anomalous Conductivity Trends: The conductivity data in Figure 7a shows counterintuitive trends. Specifically: (1) The CB-LIC matrix with 10% CB has a lower ionic conductivity than the 15% CB sample. (2) The overall trends for both ionic and electronic conductivity do not follow the expected monotonic relationships with increasing CB content. Explain the trend more.
6. Rate Capability and Data Presentation: The use of a constant current-constant voltage (CCCV) protocol makes it difficult to assess the true rate capability. The authors need to: Report the capacity contribution from the constant current (CC) phase alone for all tested rates and samples. Include corresponding charge/discharge voltage profiles.

Version 1:

Reviewer comments:

Reviewer #1

(Remarks to the Author)

I appreciate the effort the authors have made in addressing the concerns. However, I still have several significant issues regarding data analysis, particularly in the areas of cathode utilization and DEM simulation.

Q1. Regarding the response to Comment 4, the authors argue that carbon signals from adventitious sources cannot be reliably separated from those originating from carbon black (CB), and therefore, the quantification of coverage γ via XPS was conducted without considering the carbon signal. While this is a reasonable concern, I believe that such a claim should be supported with experimental evidence. Specifically, it would be helpful to present the C 1s XPS signal obtained from a control system that does not contain carbon black. This would allow readers to evaluate the baseline level of adventitious carbon and assess whether it is indeed appropriate to exclude the carbon signal entirely when quantifying coverage via XPS in this case.

Q2. It appears that the authors used a liquid-electrolyte cell as a reference system and adopted its voltage profile as a quasi-OCP. However, even at 0.02C, a finite current is still flowing, and the resulting profile cannot be regarded as a true quasi-OCP in the conventional electrochemical cell. Using a quasi-OCP to quantify the cathode utilization level in an all-solid-state cell system is good idea as demonstrated, including Ref. (<https://doi.org/10.5281/zenodo.14065128>). However, the method employed here does not yield an accurate quasi-OCP, which is precisely why I suggested performing a GITT analysis in previous revision comment (Comment 7).

If the authors insist that their 0.02C voltage profile indeed represents a quasi-OCP, then they should at the very least demonstrate that it overlaps with a GITT-derived quasi-equilibrium voltage profile. Without such a comparison, the current interpretation of cathode utilization lacks certainty

Q3. The authors state that "the objective of this study was instead to provide a coarse-grained representation of the mixing process" using DEM, and as shown in Figure R6, the DEM model appears to represent cathode particles that are already covered by LIC and carbon black. However, since the DEM simulation uses a single-property coarse-grained particle model, it does not explicitly represent the spatial distribution or interaction of individual NCM, LIC, or CB components. As a result, the simulation cannot physically capture the actual process of coating formation or quantify coating coverage in any direct manner. In this context, it remains unclear how such a simplified model can fundamentally explain variations in coating

coverage on the cathode, given that the particles are effectively assumed to be pre-coated from the outset. Is the DEM simulation merely describing the motion of already-coated particles, as shown in Figure R6? If so, how can increases in $E_{(m,sim)}$ be meaningfully correlated with progressive coating coverage? And how, in this model, is the coating process actually described? This approach appears internally inconsistent and requires further clarification.

Q4. Moreover, the manuscript draws a correlation between simulated specific energy input (based on stress intensity and stress number in Fig. 6c, d) and calculated coverage. However, the underlying relationships between these parameters and calculated coverage are not sufficiently established. The manuscript lacks a clear explanation of how coverage values were quantified in the simulation itself. I strongly encourage the authors to clarify the methodology of how coverage is calculated from DEM simulation.

Q5. Additionally, I noticed that the Young's modulus listed in Table S1 has changed from 0.055 GPa in the previous version to 0.228 GPa in the revised version. Several other Particle-Particle / Particle-Wall calibration parameters, such as static and dynamic friction coefficients, also appear to have been modified. However, no justification or documentation for these changes is provided. The authors should clearly explain why these parameters were revised and whether such changes had any effect on the simulation results shown in the figures.

Reviewer #2

(Remarks to the Author)

The authors have done an exceptional job in addressing the issues raised by the reviewers. This includes:

- new TEM results that clearly show the structures at the cathode/electrolyte interface
- better reporting of electrochemical results
- verification of assumptions in electrochemical modeling
- clarification and correction of data

Due to these improvements, I recommend that the manuscript be published with no further changes necessary.

Reviewer #3

(Remarks to the Author)

No more questions regarding this revision.

Version 2:

Reviewer comments:

Reviewer #1

(Remarks to the Author)

I thank the authors for their very detailed responses to the reviewers' comments and for the substantial improvements made to the manuscript. I believe that these revisions have significantly strengthened the paper, and that it is now suitable for publication. The study will make a very interesting and valuable contribution to the field.

Below we respond to all reviewers' comments. Changes to the manuscript are highlighted in the manuscript and the SI by yellow marks.

Reviewer #1

This manuscript, titled "Mechanofusion-derived cathode composite microstructures for solid state batteries: A scalable mixed conducting matrix coating approach," explores a promising and scalable approach for cathode composite fabrication using mechanofusion, supported by experimental studies and discrete element method (DEM) simulations. The topic is timely and potentially impactful for advancing solid-state battery (SSB) technologies.

While the proposed strategy is conceptually interesting, the current version of the manuscript raises several scientific concerns that limit its suitability for publication in Nature Communications. In particular, the interpretation of XPS coverage data lacks sufficient quantitative grounding and is not robustly linked to the presented electrochemical or structural performance metrics. In addition, some critical datasets appear either incomplete or selectively discussed, which affects the transparency and reproducibility of the conclusions. The DEM simulations, while helpful for illustrating process trends, rely on highly simplified assumptions and lack detailed justification for coarse-graining and boundary conditions, which undermines their predictive value.

Overall, despite the manuscript's relevance and the potential of the proposed method, the current form does not meet the rigor and clarity expected for this journal. I therefore do not recommend publication in Nature Communications at this time.

General remark by the authors of this manuscript: We thank the reviewer for the positive assessment of our manuscript and for indicating which aspects require improvement clarity. We have addressed below point-by-point all raised issues and marked the revised parts of the manuscript in yellow. We hope that the manuscript now meets the reviewer's expectations.

Generally, we would like to emphasize that while our approaches based on XPS analysis and DEM simulations may appear oversimplified at first glance, this perspective does not fully capture their intended scope. The XPS analysis was designed to provide a systematic, (semi-)quantitative and comparative assessment across a broad set of process parameters, rather than absolute quantitative coverage or coating thickness values, which allows meaningful trends to be identified and linked to electrochemical and structural performance. Similarly, the DEM simulations employ a coarse-graining approach as a necessary compromise to explore a wide parameter space efficiently, while still capturing key trends and mechanistic insights relevant to the processes studied. Taken together, both approaches provide robust, reproducible, and interpretable insights despite the inherent simplifications, and their limitations are now explicitly acknowledged in the manuscript.

Reviewer Comment 1: The authors report an increase in particle circularity with higher coating content (Fig. S1) or mixing intensity (Fig. 5c), interpreting this as evidence of homogeneous coating. However, given that the pristine NCM particles are irregular in shape (circularity ~0.7), a truly homogeneous and conformal coating would be expected to preserve the underlying particle morphology rather than significantly increase circularity. Could the authors clarify how
--

circularity increase can be reconciled with the concept of homogeneous coating? Is it possible that the observed trend instead reflects smoothing or thick overcoating in recessed regions, rather than uniform layer formation? In other words, can homogeneous coating be considered conceptually equivalent to increasing circularity in this context?

Answer: We thank the reviewer for this comment, which is absolutely right: a truly homogeneous coating, in the sense of an equally thick layer, should preserve the underlying particle morphology. The observed increase in circularity instead indicates that the coating covers the particles with varying thickness, consistent with the reviewer's suggestion of smoothing or thicker deposition in recessed regions. However, we do not claim that an increase in particle circularity is direct evidence of a homogeneous coating. In the discussion of Figure 5, 'homogenous' refers to the coating morphology which becomes smoother and less particulate.

We acknowledge that the original discussion could have been clearer and more rigorous, thus we have revised the corresponding paragraph to better reflect this distinction and added an explanatory figure as Figure S1e based on the reviewer's illustrative sketch.

Comment 2: In Fig. 2c, the cathode composite containing carbon black appears to exhibit lower porosity compared to the one without carbon black. Given that carbon black is typically a highly porous material, this result seems counterintuitive. Could the authors clarify the underlying reason for the observed decrease in porosity upon carbon black addition?

Answer: We agree that the lower porosity observed for the cathode composite containing carbon black may appear counterintuitive at first, given the intrinsically higher porosity of carbon black itself. However, this effect can be rationalized by mainly two aspects. First, the addition of carbon black increases the volume fraction of easily compressible material within the coating from 31.4 vol.% to 35.6 vol.% for the 80:20 vs. 80:20:3 compositions (cf. Table S1) and from 1.8 vol.% for the 99:1 to 2.2 vol.% for the 99:1:CB composition, cf. Table S1. This results in an increase in coating vol.% by about 13.2 % (80:20:0 to 80:20:3) and 19.9 % (99:1 to 99:1:CB) across the compositions. Thus, the samples with CB contain a lower vol.% of hardly compactable NCM material, leading to the slightly lower porosity. A clarification of this point has been added to the revised manuscript.

Second, it should also be noted that the calculated porosity depends on the theoretical mixed density ρ_{mix} (calculated based on the raw material densities and volume fractions cf. Table S1) and bulk density ρ_{bulk} (calculated based on experimental data recorded during powder compaction in the ZWICK material testing machine), according to the relation:

$$\varepsilon = 1 - \frac{\rho_{\text{bulk}}}{\rho_{\text{mix}}}$$

The addition of carbon black lowers the theoretical mixed density ρ_{mix} because of its low intrinsic solid density, which results in a smaller porosity ε even when ρ_{bulk} remains similar or is higher/lower compared to the composites without CB.

Comment 3: In Fig. 4b and the XPS section of the 'Methods', the authors estimate the coating coverage based on XPS data. Given that XPS provides information from only a few nanometers of the surface, could the authors clarify whether this depth is sufficient to confirm the formation of a uniform coating layer? As the coating content increases, the coating thickness may reach

several tens or even hundreds of nanometers, which could result in the coverage value approaching 1 regardless of the coating's actual uniformity. It would be helpful if the authors could elaborate on whether the XPS-derived coverage value reliably reflects coating thickness homogeneity.

Answer: We thank the reviewer for their important comment. As shown in **Figure R1** (which is Figure 4a of the original manuscript), the coverage value does not reflect coating thickness uniformity or homogeneity; instead, different coating morphologies can result in similar coverage values:

Figure R 1. This figure corresponds to Figure 4a of the manuscript.

XPS-derived coverage values are no suitable descriptor for the uniformity/homogeneity of coatings, in the sense of an equally thick coating layer. This is only possible with microscopic methods such as TEM, which are, however, suffering from limited statistics, time consuming sample preparation limiting the exploration of a broader process parameter space as done in this study. Nevertheless, the XPS-derived coverage value reliably reflects whether a coating layer, >10 nm thick, was achieved for the investigated particles. Under the experimental conditions, signals from more than 1000 NCM particles are detected, representing a statistically relevant average.

We have added more explanation on the interpretation of the coverage values to the Methods section.

Comment 4: In Fig. 4b, the coated cathode containing carbon black (CB) exhibits a higher coverage value than that without CB. However, this appears questionable, as the coverage value calculated from Equation (10) does not account for the carbon signal, even in CB-containing samples. For a more accurate estimation of coverage for CB-containing sample, (i) the coverage equation should include the XPS intensity of carbon, and (ii) to enable this, the spectra should be calibrated using an external reference sample or an added internal standard, rather than the carbon C 1s signal, which may be convoluted with the CB contribution. Could the authors comment on the reliability of the current approach and whether these modifications could be considered or justified?

Answer: We thank the reviewer for asking about the coverage calculation. We are not sure about the use of the term “calibration”, as this typically refers in XPS to the proper positioning of the energy rather than considering intensities. We believe the reviewer is commenting on the intensities of the spectral line and proper evaluation of intensities. It is true that Equation 10 (now 11) does include the carbon signal. The problem of including carbon signals into the calculation of the coverage γ is that one would always include signal intensity stemming from

adventitious carbon which cannot be distinguished from the CB contribution. Nevertheless, the current approach is reasonable for a comparison between different samples since the calculation considers the intensities of the Ni and Mn signals stemming from the NCM. These signals will be lower if they are shielded by the coating. Since the coatings with CB occupy a larger volume due to the lower density of CB, the shielding effect is stronger, leading to higher coverage values.

We have added an explanatory sentence to the methods section, why Equation 10 (now 11) does not contain the carbon signal.

Comment 5: At around 3.7 V, Ni-rich layered cathodes are known to undergo H2–H3 phase transitions. This voltage also exceeds the electrochemical stability window of Li₃InCl₆ (LIC). Is there a specific reason why 3.7 V was selected as the upper cut-off? Moreover, as the coating coverage increases, the interfacial contact area also increases, potentially promoting chemical side reactions when the cathode operates outside the stability window of LIC. If the intention was to isolate the physical effects of the coating, would it not have been more appropriate to limit the cycling voltage to below 3.7 V, where LIC is more likely to remain electrochemically stable?

Answer: We thank the reviewer for asking. The upper cut-off potential of 3.7 V vs. In/InLi is chosen in various studies (cf. [1]), independent of the solid electrolyte type, even when sulfides are used which possess a very narrow stability window. To allow at least a certain comparability with other and our own studies, we kept this upper cut-off value. Also, halide SE such as the employed LIC are generally known, in contrast to sulfide SEs such as Li₆PS₅Cl, for their high voltage stability as depicted in **Figure R2** [2]:

Figure Redacted

Figure R 2
bromides, iodides, oxides, sulfides, and nitrides. M is a metal cation at its highest common valence state. Taken from ref.[2]

For LIC in particular it was already successfully demonstrated by Jin et al. [3] and Wang et al. [4] that this material can be used as protective coating. It also should be noted that the NCM particles were already surface-modified by the supplier to increase the interfacial stability. This should further protect the LIC from significant electrochemical degradation.

We agree that the better interfacial contact may promote more interfacial degradation compared to composites with point contacts. However, we believe that such an intimate contact is necessary to enable fast charge transfer across the CAM|SE interface and thus fast cell kinetics. The samples with the matrix coating that have been tested electrochemically in our study possess a comparable interfacial contact, thus any resulting chemical side reactions are expected to be comparable as well.

Regarding the phase transition H2→H3 that NMC undergoes in the chosen potential range, the reviewer is absolutely right. However, without exploring this transition regime, associated with NCM volume changes, we would not have been able to observe cycling differences due to the varying elastic/plastic properties of the coating depending on the CB content (see discussion on Figure 8d-f, now 9d-f), which we believe is an important result of our study. In fact, by using

CCCV charging, we deliberately forced the NMC particles to undergo the phase transition to reveal differences. Furthermore, we employed NCM single crystals which are generally less prone to chemo-mechanical degradation in terms of intergranular particle cracking (cf. [5] & [6]).

Overall, we are confident that the conclusions of this work are independent of the exact upper cut-off potential.

[1] Puls et al., *Nat Energy* **2024**, 1–11. DOI: 10.1038/s41560-024-01634-3

[2] Wang et al., *Angew. Chem. Int. Ed.* 2019, 58, 8039–8043

[3] Jin et al., *Chem. Mater.* **2024**, 36 (12), 6017–6026. DOI: 10.1021/acs.chemmater.4c00515.

[4] Wang et al., *ACS Appl. Energy Mater.* 2023, 6, 3671–3681

[5] Conforto et al., *J. Electrochem. Soc.* 2021, 168 (7), 70546. DOI: 10.1149/1945-7111/ac13d2.

[6] Liu et al., *Nature communications* **2024**, 15 (1), 7970. DOI: 10.1038/s41467-024-52123-w

Comment 6: In Fig. 8a, the cycling performance does not exhibit a clear trend with respect to carbon content across the tested rates (0.1C, 0.3C, and 1C). Could the authors clarify whether this observation is reproducible? In addition, the cycling performance at 1C shows noticeable fluctuations in several data points. Could the authors comment on whether these deviations may be attributed to experimental factors, such as temperature variations or contact instability during 1C cycling?

Answer: We thank the reviewer for inquiring about the cycling data. Yes, the observation is reproducible, and we identified the following trend depending on the carbon content:

Thereby, one needs to differentiate between low and high C-rates. At low C-rates, the coating kinetics is less important but rather the (static) CAM utilization. Thus, the trend is that more CB in the matrix leads to higher capacities at low C-rate because more NCM particles are electronically connected, *i.e.*, these composite cathodes possess a higher *static* CAM utilization (cf. Figure 8b, now 9b). With increasing C-rate the kinetics of the matrix coating becomes more important. Here, coatings with more CB are inferior since the CB particles obscure the Li-ion pathways. Overall, the trend is based on a compromise between kinetics and utilization as elaborated in the manuscript when discussing Figure 8 (now 9). On the long-term, the cycling behavior is further affected by the mechanical properties as discussed in Figure 8d-f (now 9d-f).

Regarding the fluctuations during 1C cycling, we assume the reviewer refers to the following cycles (**Figure R3**):

Figure R 3. Modified Figure 8a from the original manuscript indicating fluctuations during 1C cycling.

This is a very attentive observation. We believe that these fluctuations are due to the fact that the CAM utilization was determined before the corresponding cycles. As now described in the Methods section, also acknowledging the following Comment 7, this involves charging with lower C-rates, thus it deviates from the monotonic CCCV charging before. This leads to fluctuations, which however do not affect the general trends in cycling behavior.

Comment 7: In Fig. 8b, the cathode utilization appears to have been calculated based on open-circuit potential (OCP). However, the manuscript does not provide any methodological details regarding how CAM utilization was derived from the OCP data. Could the authors specify the relaxation time applied during the OCP measurements? Furthermore, how do the authors ensure that the applied relaxation time is sufficient to reach equilibrium, such that the measured OCP accurately reflects the true state of charge (SOC)? It would also be helpful to provide the individual OCP values used in this analysis as part of the supporting data for Fig. 8b. In addition, it seems likely that a GITT-based voltage profile would be essential to estimate CAM utilization from measured OCP. However, no such GITT-related data or methodology is described in either the main text or the Supplementary Information. For clarity and reproducibility, the authors are encouraged to include both the methodological details and representative voltage profiles used for calculating CAM utilization from OCP.

Answer: We thank the reviewer for enquiring about the CAM utilization determination. In the Methods section we only referred to our previous publication [1] in which also the reference curve is provided (<https://doi.org/10.5281/zenodo.14065128>). For improved clarity and facilitating reproducibility, we now have added more experimental information, such as the relaxation times, into the Methods section and provide the used OCP values for Figure 8b (now 9b) in the Supplementary Data File. The reference curve, which was not a GITT curve but a quasi-OCP (titration) curve of a LIB with the identical CAM recorded at 0.02C, was already published alongside with our previous publication [1]. Thus, we have added the DOI of the public repository in which the interested reader will find the raw data for the reference as directly usable .txt file.

Regarding the reviewer's question concerning the relaxation time, please have a look at the following **Figure R4** which shows the potential profiles during the CAM utilization determination for two cathode compositions with good and bad kinetics:

Figure R 4. Representative CAM utilization step for the cathode composite with a) 80:20:0.2 and b) 80:20:3 composition.

In both cases, the voltage relaxation (dU/dt) at the end of the 5h relaxation step is in the range of only 1 mV/h or smaller. Thus, we consider the relaxed OCP values to be in equilibrium reflecting the state of charge with sufficient accuracy.

[1] M. Kissel et al., *Adv. Energy Mater.* **2025**, 2405405, DOI: 10.1002/aenm.202405405

Comment 8: In Table S1, it is unclear how the values for static friction, dynamic friction, yield ratio, and restitution coefficient were determined. Could the authors provide justification for the selection of these parameters? In addition, it is important to explain why the particle density was set to 2417 kg/m^3 , given that it does not correspond to either the theoretical material density, the measured bulk density or even an average of the two. A detailed rationale for this choice is needed to assess the accuracy of the DEM simulations.

Answer: We thank the reviewer very much for their thoughtful questions regarding the simulation parameters. Originally, the parameters were calibrated such that the average specific power acting on the rotor matched the experimentally measured specific power (we will elaborate on this in Comment 11). For a more physically grounded approach, however, we have now carried out additional calibration experiments and simulations.

First, we used the force–displacement curves obtained from the ZWICK compression tests and reproduced them in the simulations by manually adjusting the friction parameters, Young’s modulus, and other material properties (**Figure R5a**). This procedure allowed us to reliably capture the static properties, particularly the elastic modulus of the coarse grains.

In addition, we determined the dynamic angle of repose with a rotating cylinder and compared it with simulations, which enabled us to refine the frictional parameters (**Figure R5b**).

As can be seen from the compaction simulations, the maximum compaction force of 1,000 N can be reproduced satisfactorily. Nevertheless, discrepancies remain in the force increase at the onset of compaction, which can be attributed to several factors:

1. The coarse-grained particles are not bimodally distributed. As a result, they are more difficult to pack and compact, leading to a faster force increase than observed in the experiments.
2. In practice, deformations of the soft solid-electrolyte-carbon black matrix also occur, which are not considered in the current simulation framework.

The dynamic angle of repose was experimentally determined using a GranuDrum device and subsequently reproduced in Rocky simulations. In the experiments, the powder was rotated at 20 rpm, and a total of 60 images were recorded from which the dynamic angle was evaluated. The average value of these measurements is shown in Figure R5b and exhibits very good agreement with the simulated angle, confirming the correct calibration of the simulation parameters.

We have now added new figures to the Supplementary Information (Figure S10a,b), as well as an explanation of the calibration procedure in the Methods section.

Figure R 5. Calibration experiments and simulations to capture the coarse-grain properties and dynamics such as the young's modulus, friction coefficients and yield ratio.

With respect to the particle density, we experimentally determined two quantities. First, the true density of a coated particle with a mass ratio of 80:20:3 (exact composition based on 100%: 77.6:19.4:3), calculated from the volume fractions of NCM, Li_3InCl_6 , and CB, which was found to be **3.94 g/cm³** (a detailed table of all weighing and calculated densities can be found in the Supplementary Information, Table S1). Second, we measured the unconsolidated bulk density of this mixture, which amounted to **1.45 g/cm³**.

Each coarse-grained particle was further assigned an internal porosity of 0.39, resulting in a coarse-grain density of **2.417 g/cm³** (Figure R6). This procedure is now described in the Supplementary Information (Figure S10c), and for clarity we have also added a schematic illustration.

Figure R 6. Schematic illustration of the calculation of bulk porosity, mixed density, as well as coarse grain density, added as Figure S10 to the Supplementary Information.

To verify the correctness of this approach, we additionally compared the filling degree obtained from the GranuDrum experiments and the DEM simulations (Figure R7). Both yielded a filling level of approximately 33 %, confirming that the simulated bulk volume accurately reflects the experimental conditions, which also validates the coarse grain density of 2.417 g cm^{-3} . This figure can also be found in the Supplementary Information as Figure S10d.

Figure R 7. Comparison of the filling degree in the GranuDrum from the experiment and the simulation added as Figure S10d to the Supplementary Information.

Comment 9: The mechanical properties listed for the wall in Table S1 are also questionable. Available specifications for the Nobilta system indicate that the inner wall is typically made of tungsten carbide and the outer shell of stainless steel (SUS), chosen for their high durability at elevated rotational speeds. Yet, the values in Table S1, specifically the density ($4,510 \text{ kg/m}^3$) and Young's modulus (120 GPa), are much lower than those of either material (Tungsten carbide and SUS). For reference, tungsten carbide has a density above $15,600 \text{ kg/m}^3$ and a Young's modulus over 530 GPa , while SUS typically exceeds $7,500 \text{ kg/m}^3$ and 190 GPa , respectively. Rather, the listed mechanical property values appear closer to those of the NCM cathode material itself rather than the mixer wall. An explanation is needed for this discrepancy, and its impact on the reliability or transferability of DEM simulation should be discussed.

Answer: We thank the reviewer very much for this detailed and knowledgeable question. To clarify briefly: the rotor is not made of tungsten but of titanium alloy **3.7165 (Ti-6Al-4V, Ti Grade 5)** and the inner wall of the mixing chamber out of stainless steel. We have reconfirmed this

information with the manufacturer (Hosokawa Alpine Group, see **Figure R8**). Titanium typically exhibits a Young's modulus of approximately **120 GPa** and a density of about **4.5 g/cm³** and thus the assumed values are correct for the rotor for our experimental setup.

Figure R 8. Screenshot of email correspondence with the manufacturer Hosokawa Alpine AG of which the machine was used in this study.

However, the inner wall of the mixing device is made of stainless steel, as shown in the screenshot “Einschubrohr Mahlraum.” The material used (1.4404 stainless steel) typically exhibits a Young's modulus of approximately 200 GPa and a density of 7.85 g cm⁻³. These values were applied in the simulation of the mixing chamber but initially not included in the supporting information but now we have added this to the calibration table in the Supporting Information Table S2.

Comment 10: It appears that the DEM simulation was performed using a single-particle system, as suggested by the assignment of a uniform set of mechanical properties (e.g., Young's modulus and Poisson's ratio) for cathode powder mixture. Could the authors clarify how these mechanical properties, young's modulus and Poisson's ratio were determined in such powder mixture?

More critically, this modeling approach raises serious concerns about the validity of the simulation. The mechanofusion-based coating process fundamentally relies on shear interactions between mechanically distinct components, Li₃InCl₆, carbon black, and the NCM cathode. Treating all particles as mechanically identical disregards the essential mechanism of the coating process, in which differential deformation plays a central role. If the authors intended to realistically simulate the mechanofusion process, the mechanical properties of

each component should have been treated separately. Using a single set of mechanical parameters, as shown in Table S1, may be convenient for computational purposes but fundamentally misrepresents the physical interactions responsible for coating formation. This oversimplification undermines the credibility of the DEM results, particularly the calculated coverage, which is presented as a key outcome of the study. Without accounting for the material-specific mechanical behavior of each component, it is difficult to regard the simulation as a meaningful representation of the mechanofusion-driven coating process.

Answer: The reviewer is right, the DEM simulations were performed using coarse-grained particles, which represent an effective mixture of NCM, SE, and CB. The mechanical properties of these coarse grains were calibrated as outlined in the previous response: initially by matching the specific rotor power, and after the revision also by incorporating calibration experiments and simulations like the ZWICK compression as well as the dynamic angle of repose. This approach is an effective procedure to capture the dynamics of the coarse-grained particle mixture – although, as reviewer states, that we are not able to represent the individual primary particles of NCM, LIC and CB but just the overall powder behavior. Alone modelling the CB would increase the simulation effort tremendously.

As correctly pointed out, our approach does not aim to reproduce the micromechanical mechanofusion coating process at the original particle scale. Capturing the material-specific mechanical behavior of Li_3InCl_6 , carbon black, and NCM would require a fundamentally different modeling framework with breakage models and new models that could capture the significant plastic deformation and fusion of particles to a homogeneous layer, which is beyond the scope of the present study and could be the focus of future work. The objective of this study was instead to provide a coarse-grained representation of the mixing process that enables the quantification of the coarse-grain stress number and stress intensity as macroscopic descriptors of the mechanofusion conditions [1,2]. Both quantities can be used to calculate the simulated specific energy input which is further elaborated in our response to Comment 11. Despite its simplified nature, this approach already allows us to identify meaningful mechanistic trends, for instance, that high stress intensities are critical for coating formation and can only be compensated by exceptionally high stress numbers.

This insight is highly relevant for process optimization and scale-up. When the materials involved are not sensitive to mechanical stress, high stress intensities, achieved through high rotational speeds, can be applied to promote time-saving coating formation. Conversely, for mechanically sensitive materials, the stress intensity should be reduced by lowering the rotational speed, while the stress number can be increased by extending the process time. For example, in the case of coating at 1,000 rpm, achieving a target coverage of ca. 97.5 % would require extending the mixing duration by about two additional hours (*i.e.*, to a total of ~3 h).

We fully acknowledge the limitations of the current coarse-grained DEM model; however, we emphasize that it already provides valuable insights into the key processing conditions governing the mechanofusion coating process and establishes a physically meaningful framework for future, more detailed investigations. It should be noted that a fully resolved microscopic description of such high-intensity mixing processes is currently not computationally feasible due to the multi-scale nature of the process including 50 μm LIC particles down to 40 nm CB primary particles. Additionally, capturing local deformation and fusion phenomena at the NCM particle surface would likely require a coupled DEM–FEM approach [3], which could explicitly resolve particle–particle and particle–surface interactions including plastic deformation and interfacial bonding effects. Such coupled approaches could form the basis of highly interesting future studies, providing fundamental insights that are broadly applicable to various mechanochemical and powder-processing systems.

[1] C. Burmeister & A. Kwade, *Chemical Society Reviews*. **2013**, 7660-7667,

DOI: 10.1039/C3CS35455E

[2] R. Schlem et al., *Advanced Energy Materials*, **2021**, 2101022,

DOI: 10.1002/aenm.202101022

[3] H- Cheng et al., *Computer Methods an Applied Mechanics and Engineering*, 2023, 115651,

DOI: 10.1016/j.cma.2022.115651

Comment 11: The authors state that “the specific power obtained from the experiments was calibrated in the DEM simulation to match with the real power data.” However, as shown in Fig. 6b, the average net specific power over the 60-minute mixing period appears to be consistently higher in the simulation than in the experimental measurement. This discrepancy raises questions about the calibration procedure. Could the authors clarify how the calibration was conducted, and how such a noticeable difference in the average power values can still be considered a successful match? A more detailed explanation of the calibration process and acceptable error margin would be necessary to assess the credibility of the DEM results.

Answer: We acknowledge that our original presentation of the power data may have been misleading, as it suggested that the calibration was not successful. This was probably caused by the pronounced decrease in specific power observed at 7,500 rpm and 10,000 rpm during the experiments. This decrease results from the coating process itself as well as from centrifugation of material. Since the simulation represents only a steady-state condition, this decline is not captured in the simulation, which is why the simulated specific power is much more stable.

As noted in our reply to comment 8, the initial calibration was performed solely on the basis of the power data. In the revised version of our manuscript, we have complemented this by additional calibration experiments based on ZWICK compression and dynamic angle of repose. To better illustrate the adequacy of the experimental and simulation specific power agreement, we now present the comparison between the mean specific powers from both simulation and experiment (**Figure R9**), which demonstrates sufficient consistency. In the manuscript we also replaced the original figure from our first submission and used this figure instead.

Figure R 9. Comparison of the average experimentally determined net power input with the average simulated net power input. This figure is included as new Figure 6b in the main manuscript.

To further demonstrate the satisfying agreement between simulation and experiment, we additionally plotted the coverage as a function of the simulated specific energy input, in addition to the experimentally determined specific energy input shown in Figure 5a of the main manuscript. The simulated specific energy input was calculated according to

$$E_{m,\text{sim}} = \frac{\overline{SE}_{\text{CG}} \cdot SF_{\text{CG}}}{m_{\text{total}}} \cdot t_{\text{process}} = \frac{\overline{SE}_{\text{CG}} \cdot SN_{\text{CG}}}{m_{\text{total}}} = \overline{SI}_{\text{CG}} \cdot \overline{SN}_p$$

where $\overline{SE}_{\text{CG}}$ is the mean stressing energy, SN_{CG} is the coarse-grained stress number, m is the processed mass, SF_{CG} is the coarse grain stress frequency, $\overline{SI}_{\text{CG}}$ the mean stress intensity, \overline{SN}_p the particle related average stress number and t_{process} is the process time.

When comparing the coverage plotted versus the experimental $E_{m,\text{exp}}$ and simulated energy input $E_{m,\text{sim}}$, small deviations can be observed, but the overall trend is clearly identical. Both data sets were fitted using the fit function

$$\gamma = 1 - \frac{a}{(x+b)^c}$$

Additionally, we calculated the relative error between both fit functions according to:

$$\text{Relative error} = \left| \frac{\gamma_{\text{sim}} - \gamma_{\text{exp}}}{\gamma_{\text{exp}}} \right|$$

The maximum relative deviation was found to be 2.05 %, and within the relevant range of specific energy input, an average of approximately 0.46 %. This demonstrates that both the experimental and the simulated evaluation of the coverage evolution as a function of the simulated or experimentally determined specific energy input agree with sufficiently high accuracy and can therefore be equally used to assess the coating process. We thank the reviewer for their concern and have included a short discussion in the manuscript (Section 2.3) as well as **Figure R10 as Figure S11a,b** in the Supplementary Information.

Figure R 10. Left: Coverage as a function of the experimentally determined specific energy input (as in the main manuscript in Figure 5a) as well as the simulation-obtained specific energy input. Right: Fit function of the coverage-specific energy input dependency and the relative error of the fit functions.

Comment 12: To reduce computation time, the authors state that the particle size in the DEM simulation was coarse-grained to 300 μm , approximately 100 times larger than the actual experimental particle size. While this approach may offer practical advantages in terms of simulation efficiency, it raises serious questions about the quantitative reliability of the resulting data. According to Equation (2), particle size directly affects not only the particle mass (m_{particle}), but also the number of particles in the system, which in turn influences key outputs such as the stress frequency and stress number. A reduction in particle count due to coarse graining may distort the actual collision statistics and energy dissipation behavior that occur in the real system.

Could the authors clarify how this coarse graining was accounted for during the interpretation of DEM outputs? Were any correction factors or scaling strategies applied to bridge the gap

between the simulated and real particle size distributions? Without such corrections, it is difficult to assess the validity of the calculated stress-related metrics and their correlation to the experimental coating behavior.

Answer:

We appreciate the reviewer's detailed and highly relevant comment concerning the coarse-graining approach used in the DEM simulations. As correctly noted, the coarse-grained particles employed in this study (300 μm) are approximately two orders of magnitude larger than the primary experimental particles. This simplification inevitably reduces the total particle count and, consequently, the number of particle–particle interactions, affecting parameters such as the stress frequency SF , stress number SN and stress energy \overline{SE} .

That being said, we like to outline our considerations regarding possible scaling effects. The stress intensity \overline{SI} is normalized by particle mass and is therefore largely independent of particle size, since both particle mass and collision energy scale with f^3 as contact force scales with f^2 [1,2] and collision duration with f [1], which would lead to:

$$\overline{SI}_{CG} = \frac{\overline{SE}_0 \cdot f^3}{m_{\text{particle},0} \cdot f^3}$$

However, due to the narrow gap between the rotor and the chamber wall, the stressing energy might be slightly overestimated so that: $\overline{SE}_{CG} = \overline{SE}_0 \cdot f^3$ could not completely hold any more. To account for possible slight deviations, we now refer to the stress intensity as \overline{SI}_{CG} in the main manuscript.

Since \overline{SE}_{CG} increases approximately with particle size with f^3 , neglecting the change in collision frequency would artificially overestimate E_m . However, the collision frequency of the coarse-grained particles should decrease by roughly f^{-3} to f^{-4} [3,4], which in case of a f^{-3} scaling compensates for the increase in SE .

It should also be noted that there is still considerable debate regarding how coarse-grained DEM simulations should be properly scaled as there is no universally correct approach. In the case of high-intensity mixers, excessively large coarse-grained particles (1 mm e.g.) can interact differently within the narrow gap regions, meaning that the stress energy or stress frequency may no longer scale strictly with f^3 or $f^{-3} - f^{-4}$. In principle, the stress number could be rescaled by an empirical factor to approximate the expected absolute values. However, such a correction would merely shift the collision number to higher values without altering the relative trends that are central to the interpretation of our results. In order to make this distinction explicit, we have used a normalized parameter in the revised manuscript which is the mean stress number per particle \overline{SN}_p as defined for the product-related stress model and consistently refer to it as such throughout. This mean stress number is a normalized version of the original stress number of the coarse-grains ($SN_p = SN_{CG} \cdot \frac{m_{\text{particle},CG}}{m_{\text{total}}}$) and thus accounts for the coarse-graining of the particles. In case of the f^{-3} scaling of the collision frequency, the mean stress number per particle is thereby scaling independent.

$$\overline{SN}_p = \frac{SF_0 \cdot t_{\text{mix}} \cdot f^{-3} \cdot m_{\text{particle},0} \cdot f^3}{m_{\text{total}}}$$

With both parameters, \overline{SI}_{CG} & \overline{SN}_p , the simulated specific energy input can thus also be calculated as follows:

$$E_{m,\text{sim}} = \overline{SI}_{CG} \cdot \overline{SN}_p$$

In the end, what is most important is that the simulated and experimental trends of net power, specific energy input, and coating coverage are comparable, which they indeed are, as

demonstrated in our reply to Comment 11 and in the manuscript and Supplementary Information. The good agreement between the experimentally measured and simulated specific power confirms that the particle dynamics are captured with sufficient accuracy, even though the DEM approach necessarily represents a simplification of the real system. And this is a promising result, showing that coarse-grained DEM simulations can be used at least for practical investigations in the future.

[1] Chu et al, *Minerals Engineering*. **2016**, DOI: <https://doi.org/10.1016/j.mineng.2016.01.020>

[2] Bierwisch et al., *Journal of the Mechanics and Physics of Solids*, 2009, DOI: <https://doi.org/10.1016/j.jmps.2008.10.006>

[3] Sakai et al., *Chemical Engineering Journal*, **2014**, <https://doi.org/10.1016/j.cej.2014.01.029>

[4] De et al., *Computer Methods in Applied Mechanics and Engineering*, **2023**, <https://doi.org/10.1016/j.cma.2023.116436>

Comment 13: It would be helpful if the meaning of each factor in Equation (4) were briefly explained in the manuscript, even though the full details are provided in reference 49. Some of the terms are currently omitted. Including this information would improve the clarity and self-containment of the methodology section, particularly for readers who may not be familiar with the prior work.

Answer: We thank the reviewer very much for pointing that out. We have now added a short explanation to each factor to improve the clarity and self-containment.

Reviewer #2

The manuscript describes an interesting study that merits publication in Nature Communications. However, the level of analysis is lacking, especially for such a high-quality journal. Additionally, some assumptions are made that are not identified and that require justification. Therefore, I recommend major changes and further experiments prior to publication. Detailed comments follow.

General remark by the authors of this manuscript: We thank the reviewer for the generally positive assessment of our study. We have addressed below point-by-point all detailed comments and marked the revised parts of the manuscript in yellow.

Comment 1: "Grain boundaries existing within agglomerates of the bare NCM disappear completely (Supplementary 167 Figure S2)." I cannot see any differences in grain boundaries in the images shown. Figure S2 is nice, but Figure 2a is too small to see these features. Larger versions should be shown (e.g. in the supplemental section), so that readers can see what is going on.

Answer: We thank the reviewer for pointing out this issue. As suggested by the reviewer, we have now added an enlarged version to the supplementary information as Figure S2b to show that the intergranular boundaries are covered with coating material.

Comment 2: Cross section images should be shown to show the thickness of the coatings on the NCM particles. Moreover, it is known that the mechanofusion process can cause strain and structural damage to the surface of materials, particularly NCM. It has also been previously reported that the surface of cathode materials may become amorphous during mechanofusion processing. For this reason, TEM should be used to examine the surface of FIB cross sections, so that the cathode surface can be mapped more carefully, with electron diffraction used to detect structural changes. Moreover, the cathode/LIC interface should be examined closely (e.g. to see if any elements from the cathode become incorporated into the LIC).

Answer: We thank the reviewer for their comment. Exemplary TEM cross-sections that show the thickness of the coating are displayed in Figures 1,3, S4 and S6. Regarding strain and structural damage, we assume that the reviewer refers to Reference <http://dx.doi.org/10.1149/2.0681913jes> which is also mentioned in Comment 6. We argue that this is not directly comparable to our study, as they used polycrystalline particles that were three times larger and therefore almost ten times heavier and thus, possess ten times higher kinetic energies, and did not contain a second weaker phase during the mechanofusion process dampening the contacts. All these aspects result in a fundamentally different processing situation and are therefore not comparable to ours.

We have carried out additional TEM-based investigations on the 80:20:3 sample to investigate potential structural and interfacial damage. To this end, scanning precession electron diffraction (SPED) was performed to investigate the structure of NCM at the interface to LIC-CB coating. An area was scanned and at each scan point a diffraction pattern recorded. Due to the precession of the beam, dynamic diffraction effects are reduced, and the recorded diffraction patterns are more kinematic-like, which facilitates pattern matching for phase recognition. The pattern matching algorithm compares the recorded diffraction patterns with simulated diffraction patterns of pre-input phases. We used templates for $\text{LiNi}_{0.82}\text{Co}_{0.11}\text{Mn}_{0.7}\text{O}_2$ as the pristine layered phase (blue), NiO as a degraded rock salt phase (yellow), and Li_3InCl_6 for the coating (red). It has to be noted that the algorithm will attribute the best matching phase to each scan point regardless of the quality of the match. Therefore, we excluded matches below a

certain index value from the map. These points are depicted in black in the phase map and represent mostly amorphous regions in the coating layer or voids.

Figure R11 below shows a TEM brightfield image of a TEM lamella prepared from the coated NCM particles. To better differentiate between NCM and LIC, a TEM darkfield image is presented. Here, the NCM particles appear dark, whereas the coating layer is brighter with very bright spots, identical to the TEM images of the entire unthinned particles. In the region marked with a red square, a SPED phase map was recorded. Only the pristine layered phase and the coating layer are detected and no degraded rock salt phase is found at the NCM-LIC interface. This was moreover manually checked by examining the diffraction patterns at the interface, which did not show any signs of degradation. Based on this, we conclude that the coating process does not damage the NCM surface region.

TEM Brightfield

TEM Darkfield

SPED Phase Map

Figure R 11. TEM images and SPED phase map of the NCM-LIC interface.

We checked the chemical stability at the interface using energy-dispersive X-ray spectroscopy (EDX). The element map (**Figure R12**) shows how the CB is embedded in a LIC matrix. To check for elemental interdiffusion, a line scan across the interface was evaluated. Due to the NCM surface being slightly slanted, both materials overlap a bit in this region. However, no sign of significant element diffusion beyond this overlapping area of approximately 10-15 nm width was detected.

Figure R 12. EDX elemental map as well as corresponding line scan across the interface.

As mentioned earlier, the situation changes when large and/or polycrystalline NCM particles are used. The reason for this lies in the fact that the collision energy scales with the cube of the particle radius, due to the corresponding cubic increase in particle mass. In addition, larger particles generally exhibit a higher defect probability, and Rumpf has shown that the aggregate

strength follows a proportionality of $\sigma \propto \frac{1}{d^2}$, further emphasizing the increased fracture susceptibility of larger particles [1].

To substantiate this, we mixed polycrystalline NCM with larger particle sizes and Li_3InCl_6 in an 80:20 ratio under the most intense processing conditions applied in this study (10,000 rpm for 60 min). The results (**Figure R13**) clearly show that, although many particles retained their overall morphology, a few fragmented or fractured NCM particles were formed. Notably, these were consistently among the largest particles, confirming both our theoretical considerations and the theory of Rumpf [1].

Figure R 13. a Polycrystal NCM before processing and b after 60 min of mixing at 10,000 rpm with LIC in an 80:20 weight ratio.

Moving to a different, more application-relevant solid electrolyte such as a sulfide might lead to critical process induced degradation mechanisms in the SE phase, which we are definitely going to investigate in future degradation studies. We have added a paragraph at the end of section 2.3 to discuss the SPED and STEM-EDX results, as well as included a new Figure 7 and Figure S13.

[1] C. Schilde, A. Kwade, *J. Mater. Res.*, 27, **2012**, pp. 672-684, DOI: 10.1557/jmr.2011.440

Comment 3: A detailed analysis of the XRD diffraction patterns is needed. Especially Rietveld refinement should be performed on all sample XRD patterns (i.e. all compositions and all processing conditions) to detect any changes in lattice constants and cation mixing extent in the cathode after processing. Further analysis should be performed to derive values of the lattice strain and grain size (which is related to the defect concentration). This is highly important information for all samples, since the electrochemical performance of NCM is very sensitive to strain or the accumulation of defects. The XRD patterns shown in Figure S5 are also much too small. This figure should be made much larger, so that readers can see any changes in XRD patterns (e.g. expanding this figure to the size of an entire page would be nice).

Answer: We thank the reviewer for suggesting additional XRD measurements. It is indeed an interesting question whether the process leads to changes in the NCM particles. To investigate this, we measured XRD on the samples that were most stressed, i.e., at 10.000rpm for 60 minutes. We chose samples with a thick coating (composition of 80:20:3) and a thinner coating (composition of 95:5) and the diffractograms are depicted in **Figure R14**, which we included as new Figure S12 in the Supplementary Section.

Figure R 14. X-ray diffractograms for NCM-LiCl₆-CB composites processed under intense conditions (10 000 rpm for 60 min) as well as for a pristine reference.

The fact that the active material and Li₃InCl₆ share most positions in the diffraction pattern (cf. Figure S5) makes a precise evaluation of the micro-strain parameter difficult, since it is directly derived from a Full Width at Half Maximum (FWHM) contemplation of the diffraction peaks. The lattice parameters which we derived for the three NCM samples after Rietveld refinement are listed in Table R1 (which is included in the Supplementary Section as Table S3) together with calculated micro strain values and mass ratios. We observe that the lattice parameters between the NCM samples (*R* $\bar{3}$ *m* space group) are barely changing, besides a slight decrease in the *c*-parameter for the most coated 80:20:3 sample. This indicates that there is no cation mixing happening, which would result in changes in the lattice parameters.

Table R1: Lattice parameters, lattice strain, and mass ratios out of Rietveld refinement of three exemplary samples.

	a / Å	c / Å	Lattice strain / %	NCM / %	Li ₃ InCl ₆ / %
pristine	2.895739	14.304153	0.037	100	-
95:5	2.895918	14.304236	0.05	92	8
80:20:3	2.895848	14.303675	0.045	79	21

Regarding the lattice stain, the pristine material shows a micro strain of 0.037 %, which is increasing to 0.045 % for the 80:20:3 sample and 0.050 % for the 95:5 one. Those values indicate an increased lattice strain for the coated materials, which could be a result of processing the active material in the high-intensity ring shear mixer. With lower Li₃InCl₆ content, the thickness of the coating decreases, which in by itself absorbs kinetic energy during the mixing process. This could explain that materials with lower Li₃InCl₆ content show larger micro strain values. To validate this hypothesis, further studies should be conducted. Also, as mentioned above, Li₃InCl₆ and NCM share reflex positions, thus fitting errors can occur. Therefore, for future studies other material systems should be used to study the effect of ring shear mixing on coated NCM-samples. However, in a previous study significantly larger strain of up to 0.14 % was observed for NCM particles processed in a ball mill (cf. [1])

All those parameters indicate that there is little to no damage done to the NCM material via cation mixing or defect accumulation during processing of the material in the investigated ring layer mixer. Thus, a significant impact of structural change on electrochemical performance is not expected. Similarly, in the study the reviewer mentioned in Comment 6, no difference in the XRD patterns of the heavily damaged NCM particles compared to pristine samples had been observed. In our case, the calculated mass ratios between NCM and Li₃InCl₆ are also in line with the expected values, indicating that there is no amorphization happening. Furthermore, all

coated materials which have been tested electrochemically, contained approximately the same content of Li_3InCl_6 . Therefore, it can be expected that the micro strain is similar between the tested samples, thus it does not change the drawn conclusions of our study.

We have elaborated the discussion in our manuscript regarding potential material damage during processing, since these are certainly important aspects for future degradation-focused studies based on our proposed approach. Furthermore, we have added Table R1 and Figure R14 into the Supplementary Information as new Table S3 and Figure S12.

[1] F. Frankenberg et al., *Small*, 2025, 21 (41), e07279. DOI: 10.1002/sml.202507279.

Comment 4: "This assumption is indeed reflected in the cycling performance of the full cells depicted in Fig. 8a." All cells constructed in the manuscript cycled versus a InLi counter/reference electrode should correctly be referred to as "half-cells". (please see, for instance, <https://doi.org/10.1002/bte2.20220052> for a good discussion of the distinction between solid-state full cells and half-cells).

Answer: We thank the reviewer for the reminder on full/half-cells. We have corrected the notation to "half-cells" throughout the manuscript.

Comment 5: "As discussed in our previous work (40), this can be misleading," Indeed. However, not showing the plot in units of $\text{mAh}/(\text{g NCM})$ can be equally misleading. This is because it is very possible that the processing conditions could cause a reduction in the capacity of NCM through defects or induced strain, which is not considered here. It needs to be seen if processing conditions, reduce the capacity of NCM. Therefore, the plot should be shown both ways.

Answer: We thank the reviewer for raising this concern. We have added the plot in units of $\text{mAh}/\text{g}_{\text{NCM}}$ as Figure S15 in the Supplementary Section. However, we would like to note that both ways of plotting are not able to reflect any changes (reduction) in the NCM-intrinsic capacity under the given cycling conditions. In fact, both capacity abscissa axes are only scaled by a factor, maintaining the same trend. In both cases, no pure material properties are displayed but rather a convolution with kinetics and the microstructure, since in both cases an NCM mass is used for the calculation which does not consider inactive CAM particles. As seen in Figure S18, no significant differences between the voltage-capacity curves of the LIB and the active specific capacity curve of the SSB are observed. The visible shift is due to kinetic effects since the SSB was cycled at 0.1C while the LIB was cycled a 0.02C. To prove a reduction of the intrinsic NCM capacity one would need to perform GITT measurements with precisely known masses of the processed NCM particles. As the latter requirement is challenging for SSBs, one would need a LIB with the processed CAM, which would require a removal of the coating first adding further error sources. As mentioned in our replies to the reviewer's Comments 2&3, a detailed degradation study was beyond the scope of this work which employs a model system. Also, we expect that for all samples that were tested electrochemically, any damage should be comparable, thus this should not affect the conclusions of our work. We have revised the paragraph of cited sentence to be more precise and added further details on this matter.

Comment 6: "Static capacity losses consider the issue of incomplete CAM utilization which is substantial for the tested compositions as depicted 389 in Fig. 8b." The determination of CAM utilization and active specific capacity should be fully explained in the experimental section or in supplemental information. Currently, the reader is referred to Reference 40 for this explanation, where Reference 40 itself refers the reader to another reference. Therefore, no reference contains the complete description and the reader must go two references deep to

find out what is going on. It is important to clearly explain this procedure, since the data shown in Fig. 8b assumes that the mechanofusion processed CAM has an identical voltage curve to the pristine CAM. This assumption needs to be communicated to the reader. Furthermore, it needs to be justified that this assumption is true, even though previous studies have shown that mechanofusion processing of NCM can cause major changes to the voltage curve (e.g. see <http://dx.doi.org/10.1149/2.0681913jes>).

Answer: We thank the reviewer for pointing this out. We have added an explanation on how we experimentally determined the CAM utilization to the Methods section. We are also thankful for pointing to the reference of Zheng et al. 2023 (*J. Electrochem. Soc.*) [1]. As elaborated in our answer to Comment 2, their experiment is not directly comparable to ours.

We have verified that there are no major changes in the voltage curve as it had been observed in the reference mentioned:

Figure R 15. Comparison of the voltage curve of the LIB reference cell (0.02C) and an SSB with mechanofusion processed particles (0.1C with composition 80:20:0.5).

Figure R15 depicts the voltage curve of the pristine CAM in a LIB which was used as reference curve for the CAM utilization determination. For comparison, the voltage curve of an exemplary SSB cell of this study is shown using the gross specific capacity (charge per total mass of NCM in the cathode) and the active specific capacity (charge per mass of electrochemically active NCM particles in the cathode). Details regarding gross and active specific capacities can be found in reference [2].

As seen in Figure R15, the curve of the active specific capacity is well comparable to that of the LIB with a pristine NCM, in contrast to [1]. We only note minor differences below 3.1 V vs. In/InLi. The potential range which was used for the determination of the CAM utilization (3.1 – 3.4 V vs. In/InLi, marked in yellow), is well comparable and only shifted due to the overpotential.

We have added Figure R15 as Figure S18 to the Supplementary Information and mention the verified assumption in the Methods section. We would like to note that we are about to submit a manuscript which critically discusses the determination of the CAM utilization as done in this study. Thereby, we also considered the important prerequisite of identical voltage curves. In this context, we would like to note that the determined absolute values for CAM utilization might not be 100% exact, but since all cells of Figure 8 (now 9) experienced a basically identical processing history and are referred to the same LIB reference cell, a comparison among the tested cells and the observed trend is very valid.

[1] L. Zheng et al 2019 *J. Electrochem. Soc.* **166** A2924, DOI: 10.1149/2.0681913jes

Comment 7: Electrochemical characterization is insufficient. Full voltage curves and full differential capacity curves should be shown at the first cycle and subsequent selected cycles. A table should be provided with the following information for each composition: first reversible capacity, initial coulombic efficiency, and capacity fade/90 cycles. Plots of coulombic efficiency vs. cycle number should be shown. These properties should be discussed in relation to composition.

Answer: We have added all full voltage and dQ/dV curves for the initial cycle and last cycle (i.e. 4th cycle) at 0.1C, as well as for one cycle at 0.3C and 1C, as Figure S16 in the supplementary section. The table with the requested information is included as Table S4 in the supplementary section as well. The coulombic efficiency plot has been added as Figure S15; its evolution, especially for the samples with high CB content, follows a distinct trend. We believe this is related to the determination of the CAM utilization which interrupts the monotonic CCCV cycling. After these determination steps, the CE drops and partially recovers during the subsequent 20 cycles. More detailed, preferentially operando measurements might elucidate the underlying processes. However, the performance of the corresponding cells remains not promising. Overall, we assume a complex convolution of chemo-mechanical and cycling condition effects leading to non-straightforward behavior.

We have now elaborated the discussion of Figure 8 (now 9) supported by the new figures to improve clarity and robustness regarding the trend with respect to the composition and C-rate. We further added a Supplementary Note 4 to communicate the hypotheses regarding the Coulomb efficiencies.

Comment 8: The "known raw material densities" of each component used to calculate the pure mixing density should be stated. It should also be stated clearly if the "known raw material densities" are particle densities or bulk densities and how these densities were determined.

Answer: We thank the reviewer for that comment; we have now included the known raw material densities in the Methods section. These refer explicitly to the material densities, and not to the bulk densities. However, the bulk densities of the composite mixtures in their slightly consolidated state were determined from the ZWICK machine data, which also allowed us to calculate the porosity of the lightly compacted mixtures. The mixed densities used for the porosity calculation are provided in the Supplementary Information now (Table S1). These values were derived from the respective raw material densities and their corresponding volume fractions.

$$\rho_{\text{mix}} = \text{vol. \%}_{\text{NCM}} \cdot \rho_{\text{NCM}} + \text{vol. \%}_{\text{LIC}} \cdot \rho_{\text{LIC}} + \text{vol. \%}_{\text{CB}} \cdot \rho_{\text{CB}}$$

Comment 9: The actual material amounts (in grams) of each component used in the high intensity mixing step should be stated (e.g. in a table in the supplemental section).

Answer: The material quantities in both mass (grams) and volume (calculated from the raw material densities) are now provided in Table S1. To ensure consistent mechanical stress conditions during mixing, we maintained a constant total solid volume of approximately 4.91 cm³ (excluding pores).

Comment

10:

"First, 40 mg of LiPSCl (Argyrodite-635 CMP5 from Posco JK Solid Solution, South Korea) was pressed by hand." This statement is vague. Maybe it means that the powder was put into the cell in an even layer and then pressed by hand?

Answer: Yes, the reviewer is right. We have modified the corresponding sentence in the Methods section to improve clarity.

Comment 11: "On the LPSCI side of the separator" Please clarify what is meant by "separator".

Answer: Separator refers to the bilayer separator composed of a LIC and a LPSCI layer as depicted in Figure 8a (now 9a). We have modified the corresponding sentence in the Methods section to improve clarity.

Comment 12: "The C-rates were calculated based on a specific capacity of 200 mAh/g" This is vague. The capacity was based on 200 mAh per gram of what material?

Answer: It refers to gram of NCM. We have added this information to avoid ambiguity.

Reviewer #3

The manuscript presents a compelling mechanofusion approach for creating a mixed conducting matrix on NCM cathodes for solid-state batteries, demonstrating promising electrochemical results. The use of discrete element method (DEM) simulations is a significant strength. However, some clarification and additional data are required to fully support the conclusions.

General remark by the authors of this manuscript: We sincerely thank the reviewer for the positive assessment of our results. We have addressed below point-by-point all the issues raised and marked the revised parts of the manuscript in yellow. We hope that this now fully supports our conclusions.

Comment 1: NCM Particle Integrity and Structural Stability: The reviewer observed cracks in NCM particles after mechanofusion (Figure 1d) and is concerned about the structural stability of the cathode under the intense processing conditions (10,000 rpm/60min). Please provide a thorough explanation of how the NCM particles maintain their integrity, supported by a quantitative assessment of particle damage (e.g., wider-field SEM images) across different processing stages. The discussion should also address the mechanical properties of NCM that enable this stability and the potential impact on electrochemical performance.

Answer: We thank the reviewer for this comment. The cracks that can be observed in Figure 1d are intergranular boundaries since the single crystals obtained by the supplier are in fact not real single crystals but also exist in the form of aggregates (Supplementary Figure S2a). We have taken additional FIB-SEM cross sections (**Figure R16**) and did not observe any significant particle cracking under the most intense processing conditions (10,000 rpm for 60 min).

Figure R 16. FIB-SEM cross sections on composite after intense processing (10,000 rpm for 60 min).

We believe that there are three main reasons for the maintained particle integrity. First, the particles are single crystals (apart from the mentioned aggregates, which are not sufficiently deagglomerated after synthesis), so fracture would need to occur by cleavage along lattice planes, which is far less probable than intergranular cracking in polycrystalline NCM. Second, the softer LIC-CB matrix acts as a mechanical buffer that primarily deforms due to its lower Young's modulus, thereby absorbing a substantial part of the impact energy. Third, the single-crystalline NCM particles are significantly smaller ($d_{50} = 2 \mu\text{m}$) than typical polycrystalline ones. Consequently, their impact energy is expected to be roughly proportional to the particle mass, *i.e.*, to the cube of the particle diameter, and therefore much lower at equal relative velocity. Overall, the lower particle mass of the single crystals appears to be the dominant factor, resulting in reduced impact stress and improved mechanical stability during processing. We further have investigated potential structural damage with precession electron diffraction and

XRD measurement with subsequent Rietveld refinement, and have not found any significant evidence for structural damage. The results are now discussed in the manuscript at the end of section 2.3.

Comment 2: Comparison to Solution-Based Methods: The manuscript should explicitly compare the high-energy mechanofusion process to more common solution-based coating methods (e.g., Sun et al., Nano Energy 76 (2020) 105015). This comparison should cover: (1) Processing complexity and simplicity: How do the two approaches differ in terms of required equipment and steps? (2) Scalability and commercial viability: Is mechanofusion a practical method for large-scale production? (3) Resultant coating quality and performance: How do the final coatings and electrochemical performance compare? The authors need to justify why the energy-intensive mechanofusion process is a preferable method despite its apparent drawbacks.

Answer: We thank the reviewer for this interesting suggestion. We have added a new discussion part before the conclusion to discuss the aspects mentioned; please see there for details. To summarize the main points here: We believe that each approach (solution- or mechanofusion-based) has advantages and disadvantages and their preferability depends on the specific material system. We did not include a comparison of numbers since the study that the reviewer mentioned did use a different CAM in addition to different cycling conditions and loadings which makes a meaningful comparison of electrochemical performance impossible in our opinion. We hope that our manuscript did not make the impression that the presented mechanofusion approach is preferable without any constraints. Our goal is to offer an alternative design approach based on dry processing avoiding potential side reactions of the solid materials with the solvent. However, as known for chemical reactors also ring-shear mixers are available in large sizes (e.g. the CoriMix® Type CM ring layer mixer from Loedige Maschinenbau GmbH is designed for continuous operation and is available in various sizes as production mixer with capacities up to 500 m³/h).

Comment 3: Side Reactions and Interfacial Stability: Given that prolonged high-energy processing can exacerbate side reactions between LIC and NCM, leading to the formation of ion-electron insulating byproducts at the cathode-electrolyte interface (CEI), the authors must explicitly address how the impact of these reactions is minimized or managed.

Answer: We thank the reviewer for raising this important point. We fully agree that both prolonged processing times and high rotational speeds can promote side reactions at the cathode-electrolyte interface, potentially leading to the formation of ion-electron insulating byproducts.

Our work focuses on a model material system and aims to elucidate how the matrix composition affects the electro-chemo-mechanical properties of the composite architecture. The processing parameters used here were chosen to ensure homogeneous coating and good mechanical integration but were not specifically optimized with respect to interfacial stability, as this was beyond the scope of the present work. Nevertheless, the electrochemical performance is quite promising given that there is still room for significant process optimization.

A systematic investigation of such degradation mechanisms, however, is beyond the scope of this study. However, within the investigated parameter space, we did not observe any significant signatures of interfacial degradation, such as elemental interdiffusion (Supplementary Figure S13) or structural changes (Figure 7, Supplementary Figure S5). A dedicated degradation study would be required to fully capture and quantify such effects for application-relevant material systems.

We have clarified this aspect in the manuscript at the end of Section 2.3.

[1] Jin et al., *Chem. Mater.* **2024**, 36 (12), 6017–6026. DOI: 10.1021/acs.chemmater.4c00515.

[2] Kim et al., *Advanced Functional Materials* **2023**, 33 (12), 2211355. DOI:

10.1002/adfm.202211355.

Comment 4: Data Analysis Discrepancy (Coating Thickness): The linear relationships between coating mass and thickness shown in Figures 2b, 2c, and S3 are inconsistent with the physical properties of the materials. Since LIC and carbon black (CB) have different densities, the slope of the linear regression should change when CB is added, but the manuscript shows consistent slopes. The authors need to re-evaluate this data, account for the density differences, and provide a clear explanation for how they derived accurate and credible results.

Answer: We thank the reviewer very much for pointing this out. In fact, the trend in coating thickness for the composites containing carbon black shows a slightly different behavior, which admittedly was difficult to recognize in the original figure. To improve clarity, we have now added a linear fit to both data sets, which shows their slightly different slopes (**Figure R17**).

Figure R 17. Theoretical coating thickness depending on coating content. (Figure is part of the Supplementary Information as Figure S3)

From this fit, it becomes evident that the slope for the composite containing carbon black is slightly steeper than that of the carbon-black-free composite.

To further clarify how the coating thickness was calculated, we have included a detailed explanation in the Supplementary Information as Supplementary Note 2. In brief, the coating volume V_{coating} was first determined from the respective mass and volume fractions of the components.

$$V_{\text{coating}} = \frac{m_{\text{coating}}}{\rho_{\text{coating}}} = \frac{\text{wt. \%}_{\text{coating}} \cdot m_{\text{NCM}}}{\rho_{\text{coating}}}$$

Based on simple geometric considerations (**Figure R18**), the coating thickness s was then calculated by rearranging the corresponding volume relation.

$$V_{\text{coating}} = \frac{4}{3} \cdot \pi \cdot ((s + R_{\text{NCM}})^3 - R_{\text{NCM}}^3)$$

$$s = \sqrt[3]{\frac{V_{\text{coating}} \cdot 3}{\pi \cdot 4} + R_{\text{NCM}}^3} - R_{\text{NCM}}$$

Figure R 18. Illustration of the geometrical approach to calculate the theoretical coating thickness of the particles.

Comment 5: Anomalous Conductivity Trends: The conductivity data in Figure 7a shows counterintuitive trends. Specifically: (1) The CB-LIC matrix with 10% CB has a lower ionic conductivity than the 15% CB sample. (2) The overall trends for both ionic and electronic conductivity do not follow the expected monotonic relationships with increasing CB content. Explain the trend more.

Answer: We thank the reviewer for pointing this out. We carefully reevaluated the data and noticed some mistakes in the original calculations due to erroneous copying of data sets. We have corrected everything, acquired also new data and recreated the graph.

We consider the difference between the ionic conductivity of the 10/15% CB-LIC matrix samples to be small compared to the other samples and within the region of uncertainty. We did not show conductivity uncertainties to not overload the figure. The reviewer is right that the conductivities do not follow a strict monotonic trend. Such a trend would require that the carbon black is equally distributed in every sample. However, we assume that the CB is not equally distributed since this material tends to agglomeration and the premixing step is not fully optimized yet. Thus, linear mixing rules are not directly applicable. We believe that using carbon nano-fibers instead of carbon black will lead to even better results and we are already working on corresponding experiments also using a more promising, high performance sulfide SE.

Moreover, given that the heights as well as the ionic conductivities of the LIC and LPSCI layer are inherently error-prone, we expect that only significant differences can be revealed in this symmetric cell measurements. Apparently, the differences of matrix conductivities with CB below 5% are too small.

As we discussed in the main text and with Figure 7b, further finite-size effects need to be considered, and partial conductivity values might have a limited meaning / predictive character for the full/half-cell performance. This is in accordance with a very recently reported systematic study by Puls et al. [1] in which the authors also did not find a correlation between partial conductivities and full cell cycling performances.

We revised the discussion of the original Figure 7 (which is now Figure 8) and explained the trend more.

[1] S. Puls et al., *ACS Electrochem.* 2025. DOI: 10.1021/acselectrochem.5c00258

Comment 6: Rate Capability and Data Presentation: The use of a constant current-constant voltage (CCCV) protocol makes it difficult to assess the true rate capability. The authors need to: Report the capacity contribution from the constant current (CC) phase alone for all tested rates and samples. Include corresponding charge/discharge voltage profiles.

Answer: We thank the reviewer for this comment. We have added the charge & discharge voltage profiles as suggested as Supplementary Figure S16.

Reviewer #1 (Remarks to the Author):

I appreciate the effort the authors have made in addressing the concerns. However, I still have several significant issues regarding data analysis, particularly in the areas of cathode utilization and DEM simulation.

Comment 1: Regarding the response to Comment 4, the authors argue that carbon signals from adventitious sources cannot be reliably separated from those originating from carbon black (CB), and therefore, the quantification of coverage γ via XPS was conducted without considering the carbon signal. While this is a reasonable concern, I believe that such a claim should be supported with experimental evidence. Specifically, it would be helpful to present the C 1s XPS signal obtained from a control system that does not contain carbon black. This would allow readers to evaluate the baseline level of adventitious carbon and assess whether it is indeed appropriate to exclude the carbon signal entirely when quantifying coverage via XPS in this case.

Answer by the authors:

We thank the reviewer for the inquiry and the specific suggestion. Following this advice, we have re-evaluated our XPS data with respect to the C 1s signal. For this analysis, we used the data underlying **Figure 4b** of the manuscript, where we compare the coverage for samples with and without CB, as “control system”.

Figure R 1 shows that the C 1s intensity is higher for the sample containing CB, which is reasonable. At the same time, the sample without CB shows a non-negligible C 1s intensity.

Figure R 1. Comparison of the C 1s signal for sample without CB (80:20:0, red) and with CB (80:20:3, green).

We then re-calculated the coverage values for the different coating contents of **Figure 4b** which are depicted in **Figure R 2** together with the original values (gray squares).

In the first approximation, we used the total C 1s signal intensity (green circles), which increases the overall coverage values but maintains the same trend. In the second approximation, the corrected C 1s signal intensity (red triangles), obtained by subtracting the

C 1s intensity of the respective control system, is used for the coverage calculation. Here, the data point for the 98:2:CB composition clearly deviates from the trend. This deviation can be understood by inspecting the survey spectra in **Figure R 3**, which shows that the overall intensity (including C 1s) is higher for the sample without CB than for the one with CB. Subtracting the C 1s intensity of the CB-free control sample therefore overcorrects the C 1s signal and leads to an artificially low coverage value.

Figure R 2. Comparison of differently calculated coverage values.

Figure R 3. Comparison of the signal intensities for sample without CB (98:2:0, red) and with CB (98:2:0.3, green).

We performed an analogous analysis for the data shown in **Figure 5a** of the manuscript and made similar observations. Again, the overall intensity baseline differs from sample to sample, as illustrated in **Figure R 4**. Nevertheless, the trend of the coverage as a function of process time is preserved (**Figure R 5**) irrespective of whether the corrected C 1s signal is included in the calculation or not.

Figure R 4. Comparison of the C 1s signal for samples with composition 80:20:3 processed at 5000 rpm for different times. The red spectra represents the control sample without CB used for the recalculated coverage values of **Figure R 5**.

Figure R 5. Comparison of original and re-calculated coverage values for the 80:20:3 sample processed under various conditions.

These results indicate that using absolute C 1s intensities from a nominally CB-free control sample as a reference for recalculating the coverage is not robust. The main reason is that the total C 1s and survey intensities depend on several experimental factors that cannot be

perfectly controlled in practice and thus vary from sample to sample. This makes a quantitative comparison based on absolute C 1s intensities error-prone.

Based on this analysis, we conclude that including the C 1s intensity in the coverage calculation does not provide a more reliable measure of the coating, and we therefore retain the coverage values derived from Ni and Mn attenuation.

We have added the following explanatory sentence into the methods section: *Moreover, the overall signal intensity, including C 1s, depends on several experimental parameters and shows noticeable variations between nominally similar samples. Thus, even the corrected C 1s intensity, obtained by subtraction the C 1s intensity of a CB-free reference sample, was not used in the coverage calculation.*

Comment 2: It appears that the authors used a liquid-electrolyte cell as a reference system and adopted its voltage profile as a quasi-OCP. However, even at 0.02C, a finite current is still flowing, and the resulting profile cannot be regarded as a true quasi-OCP in the conventional electrochemical cell. Using a quasi-OCP to quantify the cathode utilization level in an all-solid-state cell system is good idea as demonstrated, including Ref. (<https://doi.org/10.5281/zenodo.14065128>). However, the method employed here does not yield an accurate quasi-OCP, which is precisely why I suggested performing a GITT analysis in previous revision comment (Comment 7).

If the authors insist that their 0.02C voltage profile indeed represents a quasi-OCP, then they should at the very least demonstrate that it overlaps with a GITT-derived quasi-equilibrium voltage profile. Without such a comparison, the current interpretation of cathode utilization lacks certainty.

Answer by the authors:

We thank the reviewer for the thoughtful comment and fully agree that the determination of a quasi-OCP requires careful consideration of current-induced overpotentials. **Figure R 6** shows a comparison of a LIB reference curve at 0.05C with the GITT-derived quasi-equilibrium voltage profile. Both profiles coincide within experimental accuracy, especially in the yellow voltage region, which was used for the CAM utilization determination, demonstrating that the low-current voltage curve used in our analysis represents a reliable quasi-OCP.

We note that the quasi-OCP reference curve used in the present work was originally recorded at 0.02C and published previously (<https://doi.org/10.5281/zenodo.14065128>), whereas the new GITT experiment was performed on a separate cell for a subsequent study using 0.05C as the low-rate condition. Since at 0.02C even smaller currents are flowing than at 0.05C, the associated overpotentials are expected to be equal or lower. The fact that the 0.05C profile already overlaps with the GITT quasi-equilibrium curve within experimental error, especially in the potential region used for the CAM utilization determination (yellow box in **Figure R 6**) therefore supports the validity of the 0.02C profile as a quasi-OCP reference.

In this context, we acknowledge that the absolute values of CAM utilization may not be perfectly exact. However, since all cells shown in **Figure 9** are referenced to the same reference cell, we argue that the relative comparison among the tested cells, including the current interpretation and observed trend in CAM utilization remains robust and scientifically meaningful.

We would also like to note that we are currently preparing a separate manuscript that critically discusses the determination of CAM utilization as applied in this work. There, we explicitly address the prerequisite of quasi-OCP conditions and matching voltage profiles for referencing and highlight the associated limitations.

Figure R 6. Comparison between the voltage versus capacity profile obtained from galvanostatic charging at 0.05C and from galvanostatic intermittent titration technique (GITT). The relaxed OCV after 2h during GITT are displayed as blue data points. The yellow box indicates the potential region used for determination of the CAM utilization.

In response to the present concern, we have included a new **Figure S18** in the revised Supplementary Information, referenced in the Method section, in which we compare the GITT-derived quasi-equilibrium voltages with the voltage profile obtained at 0.05C and believe that this satisfactorily addresses the reviewer's concern.

Comment 3: The authors state that “the objective of this study was instead to provide a coarse-grained representation of the mixing process” using DEM, and as shown in Figure R6, the DEM model appears to represent cathode particles that are already covered by LIC and carbon black. However, since the DEM simulation uses a single-property coarse-grained particle model, it does not explicitly represent the spatial distribution or interaction of individual NCM, LIC, or CB components. As a result, the simulation cannot physically capture the actual process of coating formation or quantify coating coverage in any direct manner. In this context, it remains unclear how such a simplified model can fundamentally explain variations in coating coverage on the cathode, given that the particles are effectively assumed to be pre-coated from the outset. Is the DEM simulation merely describing the motion of already-coated particles, as shown in Figure R6? If so, how can increases in $E_{(m,sim)}$ be meaningfully correlated with progressive coating coverage? And how, in this model, is the coating process actually described? This approach appears internally inconsistent and requires further clarification.

Answer by the authors:

We fully acknowledge that the DEM model, due to the use of single-property coarse-grained particles, does not capture the microscopic coating process, nor does it resolve the spatial distribution or individual interactions between NCM, LIC, and CB. Consequently, the simulation cannot, and *is not intended* to, describe the progression of coating formation or quantify

coating coverage directly. This is explicitly stated in both the previous response letter and the manuscript.

We argue that a truly mechanistic simulation of coating formation at the real particle level would require a fundamentally different modelling framework, involving dedicated breakage, adhesion, and fusion models, which to the best of our knowledge have not yet been developed for this material system. Moreover, due to the extremely high number of particles such a simulation is not possible on low micrometer and nanometer scale and, thus, such an approach is far beyond the scope of the present study. In particular, resolving the actual particle sizes would make DEM computationally infeasible. For example, explicitly representing CB with ~ 40 nm primary particles would require simulating approximately 8.95×10^{15} particles of CB alone, which is far beyond the capabilities of any current computational system. Simulating individual particle coating for a very small subset might be possible, but this would preclude capturing the macroscopic mixing process, which is essential for process understanding and design and thus not useful for our objectives.

Instead, the DEM simulation is designed to provide a coarse-grained description of the macroscopic stressing conditions in the high-intensity mixer. The mixing process is from a process point of view determined by the type of stresses, the number of particle collisions and the intensity of the collisions. In the real process, coating formation has a clear influence on the mean power input, as seen from the pronounced decrease in power over process time. The DEM does not attempt to replicate this transient evolution. Rather, it focuses on the steady-state regime, which reflects the mean power state of the process. By showing that the simulated power agrees well with the experimentally measured mean power over the whole process time, we demonstrate that the DEM model captures the overall motion and interaction of the particles with sufficient accuracy, even though it does not resolve the micro- and nanoscopic coating formation at the particle surface.

Thus, the DEM simulation does not describe the coating evolution itself. Instead, it provides statistically meaningful collision data for a representative steady state (= mean power from experiment). From this steady state, we extract the stress intensity and stress number, which together define the simulated specific energy input. Since this simulated energy input correlates strongly with the experimentally derived one due to the fact that also the mean power from the experiment matches the power from the steady state simulation, SI and SN can be used to interpret how different processing conditions influence the macroscopic energy input required to achieve a certain coating coverage. However, a certain specific energy can be achieved either by higher stress numbers and lower stress intensity or vice versa, but with different results due to different stressing conditions which can only be evaluated by the DEM simulations. The coverage is always determined experimentally (see next comment).

In summary, while the DEM model does not mechanistically simulate coating formation, it provides a validated macroscopic framework from which SI and SN can be determined. These parameters and by that the so-called stress model allow a physically meaningful correlation with experimentally measured coverage and serve as valuable tools for future process design and optimization.

We hope that this clarifies our approach. We have taken great care to make the results and restrictions of the model approach clear to the reader and think that any misunderstanding is avoided.

We have added the following explanatory sentence into the main manuscript:

While this approach does not retain information on the original coating progression, it allows for the determination of the mixer's overall stressing conditions.

Comment 4: Moreover, the manuscript draws a correlation between simulated specific energy input (based on stress intensity and stress number in Fig. 6c, d) and calculated coverage. However, the underlying relationships between these parameters and calculated coverage are not sufficiently established. The manuscript lacks a clear explanation of how coverage values were quantified in the simulation itself. I strongly encourage the authors to clarify the methodology of how coverage is calculated from DEM simulation.

Answer by the authors:

We thank the reviewer for this comment. We believe that there is a misunderstanding: The coverage was not determined from the DEM simulations. All coverage values reported in the manuscript were experimentally determined by XPS. The DEM simulations were used solely to quantify the collision behavior of the coarse-grained particles and to derive a simulated specific energy input.

Plotting the experimentally measured coverage as a function of both the simulated and experimentally measured specific energy input demonstrates good agreement between both approaches, as there is a maximum deviation of both of 2 %. This confirms that the DEM model captures the macroscopic stressing conditions of the mixing process with sufficient accuracy. More generally, correlating coverage with specific energy input, whether from experiment or simulation, allows us to determine how much energy input is required to achieve a given target coverage. The advantage of the simulation is that it enables us to deconvolute the total energy input into stress intensity and stress number, which is not accessible experimentally. This provides a direct link between stress intensity, required stress number, and thus the necessary process time to reach a desired coating coverage. In this way, the DEM framework can serve as a valuable predictive tool for future process optimization, enabling the estimation of the required stressing conditions to achieve a defined experimental target parameter value such as coverage.

Question 5: Additionally, I noticed that the Young's modulus listed in Table S1 has changed from 0.055 GPa in the previous version to 0.228 GPa in the revised version. Several other Particle-Particle / Particle-Wall calibration parameters, such as static and dynamic friction coefficients, also appear to have been modified. However, no justification or documentation for these changes is provided. The authors should clearly explain why these parameters were revised and whether such changes had any effect on the simulation results shown in the figures.

Answer by the authors: The reviewer has correctly noticed that the values in **Table S1** have changed. This is due to the fact that, following the reviewer's suggestion, additional calibration experiments were carried out to refine the DEM parameters in a more physically realistic manner. This resulted in slightly adjusted parameters, which was explained in the previous response letter (comment 8) and is provided in the revised Supporting Information of the manuscript. Importantly, the impact of these modified parameters on the simulation results was only marginal, as demonstrated by the direct before/after comparison (**Figure R7**): the stress intensity decreased slightly after parameter adjustment, but the overall trend remained unchanged. The same holds for the stress number; while the trend is identical, the normalization by particle count results in a different numerical scale.

Previous simulation

New simulation

Figure R 7. Comparison between the Coverage as a function of the stress intensity and stress number before and after calibration optimization.

This manuscript, titled "Mechanofusion-derived cathode composite microstructures for solid-state batteries: A scalable mixed conducting matrix coating approach," explores a promising and scalable approach for cathode composite fabrication using mechanofusion, supported by experimental studies and discrete element method (DEM) simulations. The topic is timely and potentially impactful for advancing solid-state battery (SSB) technologies.

While the proposed strategy is conceptually interesting, the current version of the manuscript raises several scientific concerns that limit its suitability for publication in Nature Communications. In particular, the interpretation of XPS coverage data lacks sufficient quantitative grounding and is not robustly linked to the presented electrochemical or structural performance metrics. In addition, some critical datasets appear either incomplete or selectively discussed, which affects the transparency and reproducibility of the conclusions. The DEM simulations, while helpful for illustrating process trends, rely on highly simplified assumptions and lack detailed justification for coarse-graining and boundary conditions, which undermines their predictive value.

Overall, despite the manuscript's relevance and the potential of the proposed method, the current form does not meet the rigor and clarity expected for this journal. I therefore do not recommend publication in Nature Communications at this time.

Comments

1. The authors report an increase in particle circularity with higher coating content (Fig. S1) or mixing intensity (Fig. 5c), interpreting this as evidence of homogeneous coating. However, given that the pristine NCM particles are irregular in shape (circularity ~ 0.7), a truly homogeneous and conformal coating would be expected to preserve the underlying particle morphology rather than significantly increase circularity. Could the authors clarify how circularity increase can be reconciled with the concept of homogeneous coating? Is it possible that the observed trend instead reflects smoothing or thick overcoating in recessed regions, rather than uniform layer formation? In other words, can homogeneous coating be considered conceptually equivalent to increasing circularity in this context?

2. In Fig. 2c, the cathode composite containing carbon black appears to exhibit lower porosity compared to the one without carbon black. Given that carbon black is typically a highly porous material, this result seems counterintuitive. Could the authors clarify the underlying reason for the observed decrease in porosity upon carbon black addition?

3. In Fig. 4b and the XPS section of the 'Methods', the authors estimate the coating coverage based on XPS data. Given that XPS provides information from only a few nanometers of the surface, could the authors clarify whether this depth is sufficient to confirm the formation of a

uniform coating layer? As the coating content increases, the coating thickness may reach several tens or even hundreds of nanometers, which could result in the coverage value approaching 1 regardless of the coating's actual uniformity. It would be helpful if the authors could elaborate on whether the XPS-derived coverage value reliably reflects coating thickness homogeneity.

4. In Fig. 4b, the coated cathode containing carbon black (CB) exhibits a higher coverage value than that without CB. However, this appears questionable, as the coverage value calculated from Equation (10) does not account for the carbon signal, even in CB-containing samples. For a more accurate estimation of coverage for CB-containing sample, (i) the coverage equation should include the XPS intensity of carbon, and (ii) to enable this, the spectra should be calibrated using an external reference sample or an added internal standard, rather than the carbon C 1s signal, which may be convoluted with the CB contribution. Could the authors comment on the reliability of the current approach and whether these modifications could be considered or justified?

5. At around 3.7 V, Ni-rich layered cathodes are known to undergo H2–H3 phase transitions. This voltage also exceeds the electrochemical stability window of Li₃InCl₆ (LIC). Is there a specific reason why 3.7 V was selected as the upper cut-off? Moreover, as the coating coverage increases, the interfacial contact area also increases, potentially promoting chemical side reactions when the cathode operates outside the stability window of LIC. If the intention was to isolate the physical effects of the coating, would it not have been more appropriate to limit the cycling voltage to below 3.7 V, where LIC is more likely to remain electrochemically stable?

6. In Fig. 8a, the cycling performance does not exhibit a clear trend with respect to carbon content across the tested rates (0.1C, 0.3C, and 1C). Could the authors clarify whether this observation is reproducible? In addition, the cycling performance at 1C shows noticeable fluctuations in several data points. Could the authors comment on whether these deviations may be attributed to experimental factors, such as temperature variations or contact instability during 1C cycling?

7. In Fig. 8b, the cathode utilization appears to have been calculated based on open-circuit potential (OCP). However, the manuscript does not provide any methodological details regarding how CAM utilization was derived from the OCP data. Could the authors specify the relaxation time applied during the OCP measurements? Furthermore, how do the authors ensure that the applied relaxation time is sufficient to reach equilibrium, such that the measured OCP accurately reflects the true state of charge (SOC)? It would also be helpful to provide the individual OCP values used in this analysis as part of the supporting data for Fig. 8b.

In addition, it seems likely that a GITT-based voltage profile would be essential to estimate CAM utilization from measured OCP. However, no such GITT-related data or methodology is described in either the main text or the Supplementary Information. For clarity and reproducibility, the authors are encouraged to include both the methodological details and

representative voltage profiles used for calculating CAM utilization from OCP.

8. In Table S1, it is unclear how the values for static friction, dynamic friction, yield ratio, and restitution coefficient were determined. Could the authors provide justification for the selection of these parameters? In addition, it is important to explain why the particle density was set to 2417 kg/m^3 , given that it does not correspond to either the theoretical material density, the measured bulk density or even an average of the two. A detailed rationale for this choice is needed to assess the accuracy of the DEM simulations.

9. The mechanical properties listed for the wall in Table S1 are also questionable. Available specifications for the Nobilta system indicate that the inner wall is typically made of tungsten carbide and the outer shell of stainless steel (SUS), chosen for their high durability at elevated rotational speeds. Yet, the values in Table S1, specifically the density ($4,510 \text{ kg/m}^3$) and Young's modulus (120 GPa), are much lower than those of either material (Tungsten carbide and SUS). For reference, tungsten carbide has a density above $15,600 \text{ kg/m}^3$ and a Young's modulus over 530 GPa, while SUS typically exceeds $7,500 \text{ kg/m}^3$ and 190 GPa, respectively. Rather, the listed mechanical property values appear closer to those of the NCM cathode material itself rather than the mixer wall. An explanation is needed for this discrepancy, and its impact on the reliability or transferability of DEM simulation should be discussed.

10. It appears that the DEM simulation was performed using a single-particle system, as suggested by the assignment of a uniform set of mechanical properties (e.g., Young's modulus and Poisson's ratio) for cathode powder mixture. Could the authors clarify how these mechanical properties, young's modulus and Poisson's ratio were determined in such powder mixture?

More critically, this modeling approach raises serious concerns about the validity of the simulation. The mechanofusion-based coating process fundamentally relies on shear interactions between mechanically distinct components, Li_3InCl_6 , carbon black, and the NCM cathode. Treating all particles as mechanically identical disregards the essential mechanism of the coating process, in which differential deformation plays a central role. If the authors intended to realistically simulate the mechanofusion process, the mechanical properties of each component should have been treated separately. Using a single set of mechanical parameters, as shown in Table S1, may be convenient for computational purposes but fundamentally misrepresents the physical interactions responsible for coating formation. This oversimplification undermines the credibility of the DEM results, particularly the calculated coverage, which is presented as a key outcome of the study. Without accounting for the material-specific mechanical behavior of each component, it is difficult to regard the simulation as a meaningful representation of the mechanofusion-driven coating process.

11. The authors state that "the specific power obtained from the experiments was calibrated in the DEM simulation to match with the real power data." However, as shown in Fig. 6b, the average net specific power over the 60-minute mixing period appears to be consistently higher in the simulation than in the experimental measurement. This discrepancy raises questions about the calibration procedure. Could the authors clarify how the calibration was

conducted, and how such a noticeable difference in the average power values can still be considered a successful match? A more detailed explanation of the calibration process and acceptable error margin would be necessary to assess the credibility of the DEM results.

12. To reduce computation time, the authors state that the particle size in the DEM simulation was coarse-grained to 300 μm , approximately 100 times larger than the actual experimental particle size. While this approach may offer practical advantages in terms of simulation efficiency, it raises serious questions about the quantitative reliability of the resulting data.

According to Equation (2), particle size directly affects not only the particle mass (m_{particle}), but also the number of particles in the system, which in turn influences key outputs such as the stress frequency and stress number. A reduction in particle count due to coarse graining may distort the actual collision statistics and energy dissipation behavior that occur in the real system.

Could the authors clarify how this coarse graining was accounted for during the interpretation of DEM outputs? Were any correction factors or scaling strategies applied to bridge the gap between the simulated and real particle size distributions? Without such corrections, it is difficult to assess the validity of the calculated stress-related metrics and their correlation to the experimental coating behavior.

13. It would be helpful if the meaning of each factor in Equation (4) were briefly explained in the manuscript, even though the full details are provided in reference 49. Some of the terms are currently omitted. Including this information would improve the clarity and self-containment of the methodology section, particularly for readers who may not be familiar with the prior work.

$$\overline{SI}_i^{\text{diss}} = \frac{P_i^{\text{diss}}}{c_i \cdot m_{\text{particle}}} = \frac{\sum_{c=1}^{N_{c,i}} W_c^{\text{diss}}}{\Delta t_{\text{out}}} \cdot \frac{\Delta t_{\text{out}}}{N_{c,i}} \cdot \frac{1}{m_{\text{particle}}}$$